# Adaptor linked K63 di-ubiquitin activates Nedd4/Rsp5 E3 ligase

Lu Zhu[1,2]*[†], Qing Zhang[1,2], Ciro D Cordeiro[1,2], Sudeep Banjade[1,2], Richa Sardana[1,2], Yuxin Mao[1,2], Scott D Emr[1,2]*

[1]Weill Institute of Cell and Molecular Biology, Cornell University, Ithaca, United States; [2]Department of Molecular Biology and Genetics, Cornell University, Ithaca, United States

**Abstract** Nedd4/Rsp5 family E3 ligases mediate numerous cellular processes, many of which require the E3 ligase to interact with PY motif containing adaptor proteins. Several arrestin-related trafficking adaptors (ARTs) of Rsp5 were self-ubiquitinated for activation, but the regulation mechanism remains elusive. Remarkably, we demonstrate that Art1, Art4, and Art5 undergo K63-linked di-ubiquitination by Rsp5. This modification enhances the plasma membrane recruitment of Rsp5 by Art1 or Art5 upon substrate induction, required for cargo protein ubiquitination. In agreement with these observations, we find that di-ubiquitin strengthens the interaction between the pombe orthologs of Rsp5 and Art1, Pub1, and Any1. Furthermore, we discover that the homologous to E6AP C-terminus (HECT) domain exosite protects the K63-linked di-ubiquitin on the adaptors from cleavage by the deubiquitination enzyme Ubp2. Together, our study uncovers a novel ubiquitination modification implemented by Rsp5 adaptor proteins, underscoring the regulatory mechanism of how adaptor proteins control the recruitment, and activity of Rsp5 for the turnover of membrane proteins.

\*For correspondence:
lz232@cornell.edu (LZ);
sde26@cornell.edu (SDE)

Present address: [†]Janssen Research and Development, LLC., Spring House, PA, United States

## Editor's evaluation

This paper provides an important advance in the understanding of the role of Art protein ubiquitylation by Rsp5 in the regulation of cargo import by endocytosis. Rsp5 promotes K63 linked ubiquitylation of Art proteins, thereby enhancing plasma membrane recruitment. Along with detailed biochemical analysis, this study uncovers a novel ubiquitination modification implemented by Rsp5 adaptor proteins, underscoring the regulatory mechanism of how adaptor proteins control the recruitment and activity of Rsp5 for the turnover of membrane proteins.

## Introduction

The Nedd4/Rsp5 family E3 ligases are responsible for membrane protein ubiquitination, required for endocytosis and lysosome-dependent protein degradation. Tryptophan-tryptophan (WW) domains of Nedd4 family E3 ligases bind to substrate proteins via interaction with PY motifs containing a consensus sequence P/L-P-x-Y (*Rotin and Kumar, 2009*; *Schild et al., 1996*). Other substrates lack PY motifs and instead rely on interactions with adaptor proteins that recruit the Nedd4 E3 ligase to them, exemplified by a family of arrestin-related trafficking adaptors (ARTs) that bridge the association between substrates and Rsp5 for ubiquitination (*Lin et al., 2008*). Additionally, Rsp5 adaptors include a diverse group of proteins to mediate degradation of membrane proteins localized at the PM, Golgi, endosome, and vacuole membrane (*Alvaro et al., 2014*; *Becuwe et al., 2012*; *Hatakeyama et al., 2010*; *Hettema et al., 2004*; *Hovsepian et al., 2018*; *Léon et al., 2008*; *Li et al., 2015*; *MacDonald*

*et al., 2012*; *Nikko and Pelham, 2009*; *O'Donnell et al., 2013*; *Sardana et al., 2019*; *Zhu et al., 2020*).

Many of the Nedd4/Rsp5 adaptor proteins undergo self-ubiquitination. The ART proteins Art1, Art4, and Art8 require specific ubiquitination by Rsp5 to reach full activity (*Becuwe et al., 2012*; *Hovsepian et al., 2017*; *Lin et al., 2008*). Ubiquitination of Nedd4 adaptor protein *Commissureless* is required to downregulate the Robo receptor at the cell surface of axons, essential for midline crossing (*Ing et al., 2007*; *Myat et al., 2002*). The N-lobe region of the Nedd4/Rsp5 family E3 ligase HECT domain contains an exosite which binds ubiquitin (Ub) and has been shown to orient the Ub chain to promote conjugation of the next Ub molecule of the growing polyubiquitin chain (*Kim et al., 2011*; *Maspero et al., 2011*). It was proposed that ubiquitinated Rsp5 adaptors are more active when locked onto Rsp5 but less active when unlocked by Ubp2 (*MacDonald et al., 2020*). However, the mechanism of how Nedd4/Rsp5 adaptor ubiquitination helps enhance E3 ligase function remains unclear.

In this study, we decoded the activation mechanism of how adaptor protein ubiquitination enhances E3 ligase function and how this ubiquitination itself is regulated by the deubiquitination (DUB) enzyme Ubp2. Remarkably, we discovered that the Rsp5 adaptors Art1, Art4, and Art5 are conjugated with K63-linked di-ubiquitin (di-Ub) at specific ubiquitination sites. Ubiquitination of Art5 and Art1 enhances Rsp5 recruitment to the plasma membrane (PM) thereby promoting substrate ubiquitination. Our analysis of the binding affinity of di-Ub or isolated PY motifs to Rsp5 targeted domains uncovered that K63-linked di-Ub conjugation to the adaptor protein Any1 enhances its binding to E3 ligase Pub1. Strikingly, we found that deletion of *UBP2* rescues the DUB of adaptor proteins Art5 and Art1 in the *rsp5*-exosite mutant. Our data reveal the interplay between Ubp2 and 'Rsp5 exosite engagement' to modulate adaptor protein ubiquitination. Taken together, these results serve as a portal for future studies of Nedd4/Rsp5 adaptor proteins in general.

## Results

### Rsp5 adaptor protein Art5 undergoes K63-linked di-ubiquitination

In yeast, 14 α-arrestin domain containing proteins have been identified: Art1–Art10 (*Lin et al., 2008*; *Nikko and Pelham, 2009*), Bul1–Bul3 (*Yashiroda et al., 1996*), and Spo23 (*Aubry and Klein, 2013*). These proteins have clear arrestin sequence signatures and contain multiple PY motifs that specifically interact with the WW domains in Rsp5 (*Baile et al., 2019*), and can recruit Rsp5 to specific intracellular locations. This interaction not only results in ubiquitination of cargo proteins, but also ubiquitination of ARTs themselves. In fact, several α-arrestin domain containing proteins have been shown to be ubiquitinated by Rsp5, including Bul1, Bul2, Art1, Art4, Art5, Art6, and Art8. Among these, Art5 contains an α-arrestin domain and three C-terminal PY motifs (*Figure 1A*). It has been shown that Art5 is the only ART protein required for the inositol-induced endocytosis and degradation of the PM inositol transporter Itr1 (*Nikko and Pelham, 2009*).

We found that at steady state, endogenous Art5 migrates in two major bands by SDS-PAGE, corresponding to the ubiquitinated and non-ubiquitinated species (*Figure 1B*, lane2). Mass spectrometry has previously indicated that Ub is mainly conjugated on the K364 residue of the Art5 α-arrestin domain (*Swaney et al., 2013*). We confirmed that Art5 ubiquitination was decreased by mutating K364 (*Figure 1B*, lane 3), and is completely abolished in the $art5^{\Delta PY}$ mutant in which all three PY motifs (*Figure 1A*) were mutated (lane 4), demonstrating that Art5 ubiquitination depends on its interaction with Rsp5 via PY motifs. There is a minor amount of PY motif-dependent Art5 higher molecular weight species visible on the gel (lane 2), probably due to ubiquitination on other lysines. Strikingly, the molecular weight difference (~20 KDa) between the non-ubiquitinated and ubiquitinated forms of Art5 appears to be more than one single Ub (~9 KDa), suggesting more than one Ub molecule is conjugated to the Art5 protein. To test this hypothesis, we fused the C-terminus of $art5^{\Delta PY}$ with 1, 2, 3, or 4 Ub molecules to create $art5^{\Delta PY}$-1xUb, $art5^{\Delta PY}$-2xUb, $art5^{\Delta PY}$-3xUb, and $art5^{\Delta PY}$-4xUb, respectively. Remarkably, the ubiquitinated $Art5^{WT}$ runs in line with $art5^{\Delta PY}$-2xUb, indicating that Art5 is di-ubiquitinated mainly at the K-364 residue (*Figure 1B*, lane 4–6).

We next asked what the linkage in the di-Ub is conjugated to Art5. Rsp5 mainly catalyzed K63-linked Ub chain synthesis in vivo and in vitro (*Lauwers et al., 2009*; *Saeki et al., 2009*). We therefore decided to examine whether the di-Ub moiety on Art5 is K63-linked. To test it, we analyzed the migration of $Art5^{WT}$, $art5^{K364R}$, and $art5^{\Delta PY}$ proteins in yeast strains expressing Ub-WT and Ub-K63R.

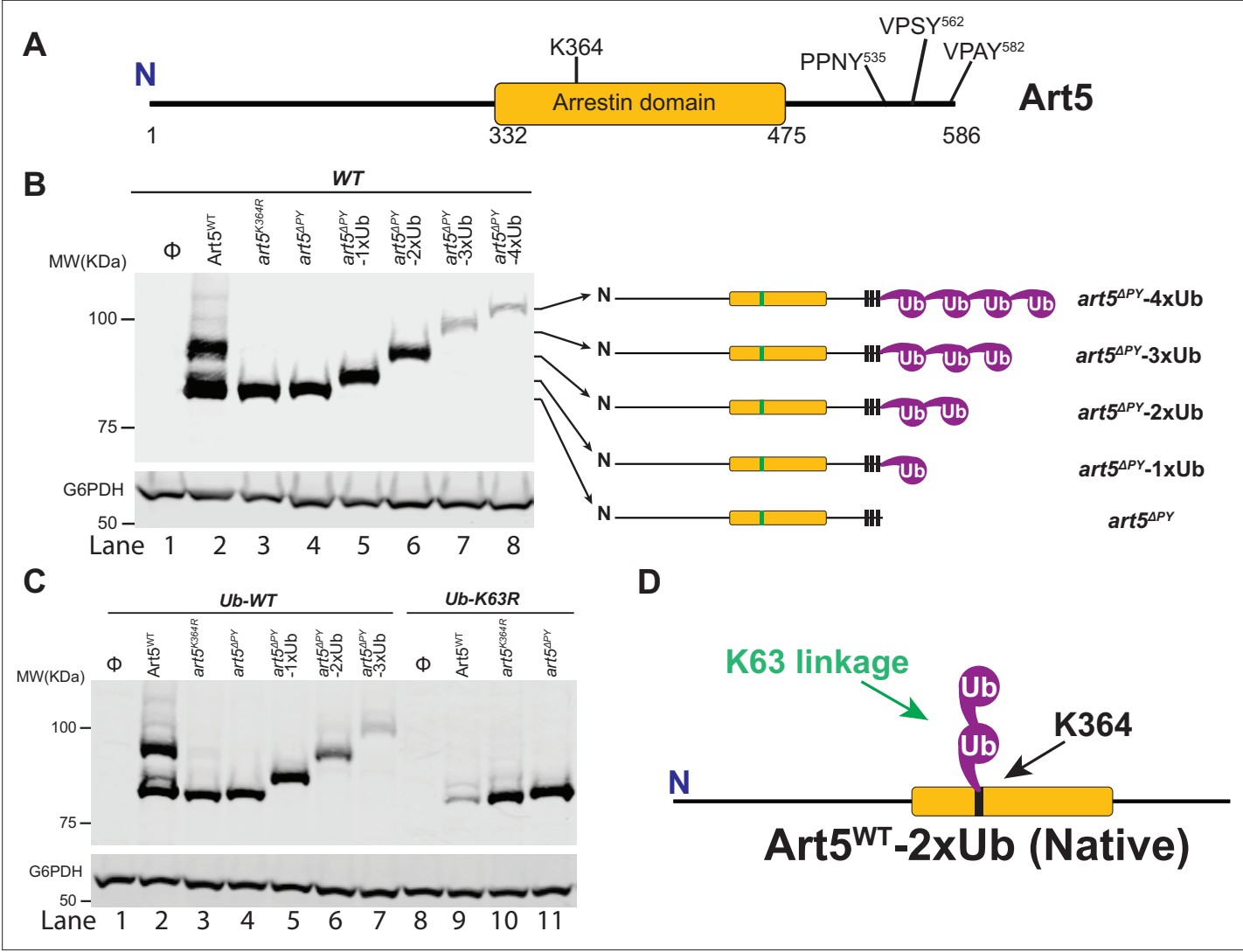

**Figure 1.** Art5 undergoes K63-linked di-ubiquitination. (**A**) Schematic representation of the domain architecture of Art5. (**B**) A di-ubiquitin (di-Ub) is conjugated at K364 residue of Art5. Western blot analysis of Art5, *art5^K364R*, *art5^ΔPY*, *art5^ΔPY*-1xUb, *art5^ΔPY*-2xUb, *art5^ΔPY*-3xUb, and *art5^ΔPY*-4xUb in the wild-type (WT) strain. (**C**) Art5 is di-ubiquitinated in a K63 linkage at the residue K364. Western blot analysis of Art5, *art5^K364R*, *art5^ΔPY* in both the *Ub-WT* and *Ub-K63R* mutant strains. (**D**) Model depicting the K63-linked di-ubiquitination of Art5 at the K364 residue. The lanes with Φ symbol indicate negative controls of empty vector. The whole cell lysate protein samples were resolved on 7% SDS-PAGE gels and the blot was probed with FLAG and GAPDH antibodies.

The online version of this article includes the following source data and figure supplement(s) for figure 1:

Source data 1. *Figure 1B, C*.

Figure supplement 1. Art1 undergoes K63-linked di-ubiquitination.

Figure supplement 1—source data 1. *Figure 1—figure supplement 1B, C*.

Figure supplement 2. Art4 undergoes K63-linked di-ubiquitination.

Figure supplement 2—source data 1. *Figure 1—figure supplement 2B*.

Figure supplement 3. UbiCRest analysis of Art1 and K63-linked di-ubiquitin (di-Ub).

Figure supplement 3—source data 1. *Figure 1—figure supplement 3A, B*.

Notably, we found that the size of the di-ubiquitinated Art5 band (lanes 2 and 3) is reduced to the mono-ubiquitinated band (lanes 9 and 10), in line with *art5^ΔPY*-1xUb (*Figure 1C*). As expected, this mono-Ubiquitin (mono-Ub) was conjugated to K364 residue. We noticed the loss of Art5 protein in the Ub-K63R mutant in a K364 and PY motif dependent manner (*Figure 1C*, lane 9), which will

be discussed later. In addition, the mono-ubiquitinated band of Art5 in the yeast Ub-K63R mutant is K364 residue dependent, confirming that the mono-Ub is conjugated mainly at the K364 residue (*Figure 1C*). Together, our results demonstrate that Art5 protein is di-ubiquitinated at the residue K364 in a K63 linkage by Rsp5.

Besides Art5, we next addressed if the K63-linked di-Ub also apply to ART-family members Art1 and Art4 (*Becuwe et al., 2012*; *Lin et al., 2008*). Art1 contains an N-terminal arrestin fold with PY motifs near its C-terminus and K486 residue is required for Art1 ubiquitination (*Figure 1—figure supplement 1A*). The ubiquitinated form of Art1 shows the same mobility shift in comparison with *art1*$^{ΔPY}$-2xUb (*Figure 1—figure supplement 1B*). The ubiquitinated band of Art1 migrates with *art1*$^{ΔPY}$-2xUb in the Ub-WT strain, while Art1 is mono-ubiquitinated at K486 in the yeast strain bearing Ub-K63R (*Figure 1—figure supplement 1C*). Art4 interacts with Rsp5 via PY motifs and can be ubiquitinated at a cluster of lysines (235, 245, 264, and 267) in the N-terminal arrestin domain (*Figure 1—figure supplement 2A*). Due to Art4 phosphorylation when cells were grown in lactate medium, Art4 protein was treated with phosphatase after being shifted to glucose containing culture medium. The ubiquitinated form of Art4$^{WT}$ migrates with the *art4*$^{ΔPY}$-2xUb and Art4$^{WT}$ was only mono-ubiquitinated in Ub-K63R condition (*Figure 1—figure supplement 2B*).

To directly determine the linkage composition of the di-Ub chain linked to Rsp5 adaptor proteins, we carried out a UbiCRest assay (*Hospenthal et al., 2015*) on Art1 K63-linked di-Ub. The Art1 protein was first immunoprecipitated (IPed) from WT cells then treated with DUBs: USP2, OTUB1, YOD1, AMSH, OTULIN, and Cezanne, respectively (*Figure 1—figure supplement 3A*). The resulting reaction mixture was resolved by SDS-PAGE gel and the bands were visualized by Western blot. The disappearance of higher molecular weight species of Art1 upon USP2 (non-specific DUB) and AMSH (K63-specific) treatment indicates that the substrate protein Art1 was ubiquitinated with a K63-linked Ub chain. Intriguingly, Cezanne treatment leads to the better release of the distal-end Ub than the proximal-end Ub from Art1-di-Ub. This result is consistent with previously published results demonstrating that Cezanne removes K63-linked Ub chains, albeit less efficient than hydrolyzing K11-linked Ub chains (*Enesa et al., 2008*; *Mevissen et al., 2013*; *Wang et al., 2017*). In contrast, no cleavage of Art1-di-Ub is visualized when incubated with YOD1, which cleaves K6, K11, K27, K29, or K33 linkage Ub chains (*Mevissen et al., 2013*). Furthermore, no cleavage of Art1-di-Ub is observed when treated with K48-specific DUB OTUB1 (*Mevissen et al., 2013*; *Wang et al., 2009*) and M1-specific DUB OTULIN (*Keusekotten et al., 2013*). In line with the UbiCRest profiling result of Art1 protein (*Figure 1—figure supplement 3A*), mono-Ub can be released when K63-linked di-Ub is treated with USP2, AMSH, or Cezanne (*Figure 1—figure supplement 3B*). Taken together, our results demonstrated that α-arrestin domain containing adaptor proteins Art1, Art4, and Art5 are di-ubiquitinated and the di-Ub is K63-linked (*Figure 1D*, *Figure 1—figure supplement 1D* and *Figure 1—figure supplement 2C*).

## Ubiquitination of Art5 is required for cargo protein Itr1 ubiquitination

We therefore sought to investigate how Art5 ubiquitination affects efficient inositol-dependent endocytosis and protein degradation of Itr1. To do so, we expressed Art5$^{WT}$ and *art5*$^{K364R}$ in an *art5*Δ mutant bearing a chromosomal Itr1-GFP. After GFP-tagged cargo proteins are sorted into the yeast vacuole, GFP is clipped from the full-length cargo proteins by vacuolar proteases (*Dupré and Haguenauer-Tsapis, 2001*; *Hettema et al., 2004*; *Jones et al., 1982*; *Kanki and Klionsky, 2008*; *Li et al., 2015*). The quantification of cargo proteins degradation efficiency is described in the section of 'Material and methods'. We found that higher inositol concentrations applied for the same amount of time results in more Itr1-GFP degradation in the WT cells and protein sorting into the vacuole lumen (*Figure 2A–C*) in an Art5 dependent manner. The *art5*$^{K364R}$ caused a severe decrease in the rate of Itr1-GFP degradation (*Figure 2A and C*) and protein endocytosis (*Figure 2B*) compared with Art5$^{WT}$. Thus, Art5 ubiquitination is required to promote efficient Itr1 endocytosis and protein degradation upon inositol-treatment.

We hypothesize that the Itr1 sorting defect in *art5*$^{K364R}$ is due to defective Itr1 ubiquitination. To test it, we expressed Itr1-GFP in a *doa4*Δ mutant bearing a Myc-Ub expression vector to stabilize ubiquitinated membrane proteins after multivesicular body sorting into the vacuole. After inositol treatment, Itr1-GFP was IPed from cell lysates prepared from yeast expressing Art5$^{WT}$ and *art5*$^{K364R}$. The ubiquitinated pool of Itr1-GFP can be detected in the Art5$^{WT}$ condition, whereas this ubiquitination was

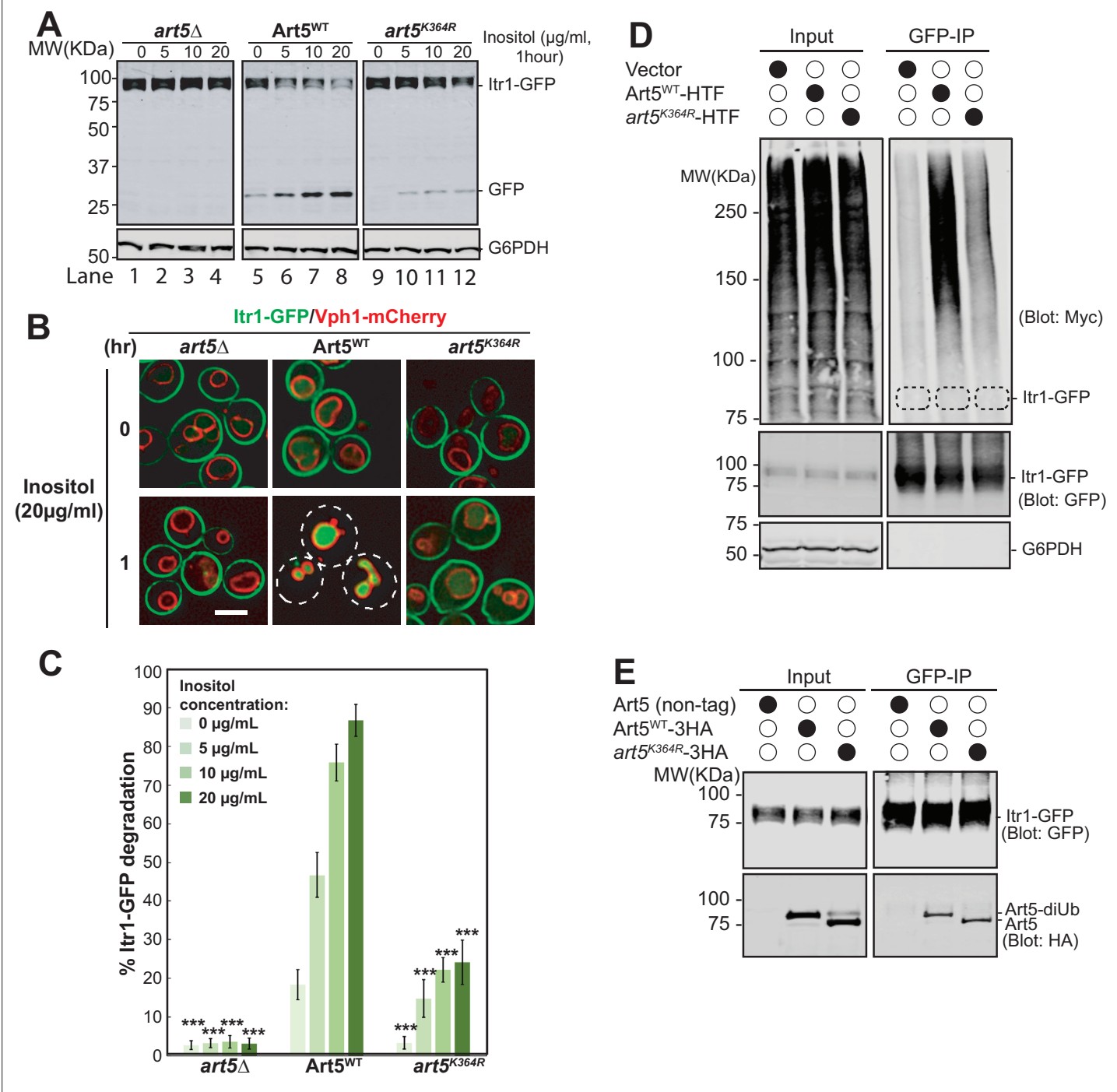

**Figure 2.** Ubiquitinated Art5 promotes cargo protein Itr1 ubiquitination. (**A**) Immunoblot analysis of Itr1-GFP degradation induced with indicated concentration of inositol for 60 min. (**B**) Fluorescence microscopy of *art5Δ*, Art5[WT], or *art5[K364R]* cells expressing Itr1-GFP and vacuole membrane marker Vph1-mCherry with or without inducing endocytosis by treating with serial dilution of inositol. Scale bar = 2 µm. (**C**) Band densities of blots in (**A**) were quantified and expressed as the mean% Itr1-GFP degradation. Error bars indicate 95% CI, n=3. ***, p<0.005 vs Itr1-GFP degradation in Art5[WT] at different inositol concentrations. (**D**) *doa4Δpep4Δart5Δ* cells expressing Itr1-GFP and Art5[WT] or *art5[K364R]* were grown to mid-log phase in synthetic medium at 30°C. Cells were pretreated with 0.1 µM CuCl$_2$ for 4 hr to induce the Myc-Ub expression before treated with 20 µg/mL of inositol. Cells were collected before and after 15 min of inositol treatment. Itr1-GFP was immunoprecipitated by GFP-Trap nanobody resin. The empty strain (*doa4Δpep4Δart5Δ*) is used as a negative control here. The dashed circles highlight the positions of non-ubiquitinated Itr1-GFP. The whole cell lysate proteins in the left gels represent the loading control and the co-immunoprecipitated protein samples were resolved in right gels.

*Figure 2 continued on next page*

*Figure 2 continued*

(**E**) Immunoprecipitation (IP) of Itr1-GFP and blotting for Art5$^{WT}$ or *art5*$^{K364R}$. Whole cell lysate and the IP reaction was resolved on 10% SDS-PAGE gels and the blots were probed with both GFP and Myc antibodies.

The online version of this article includes the following source data and figure supplement(s) for figure 2:

**Source data 1.** *Figure 2A, D and E*.

**Source data 2.** *Figure 2C*.

**Figure supplement 1.** Ubiquitinated Art1 is required for efficient Mup1 ubiquitination.

**Figure supplement 1—source data 1.** *Figure 2—figure supplement 1A, D, E, F*.

**Figure supplement 1—source data 2.** *Figure 2—figure supplement 1C*.

**Figure supplement 2.** The Npr1-mediated phosphorylation of Art1 is independent of the ubiquitination status of Art1.

**Figure supplement 2—source data 1.** *Figure 2—figure supplement 2A, B*.

**Figure supplement 3.** 2xubiquitin (2xUb) fusion with *art5*$^{K364R}$ or *art5*$^{ΔPY}$ mutants do not rescues their cargo protein Itr1 sorting defects.

**Figure supplement 3—source data 1.** *Figure 2—figure supplement 3A and C*.

**Figure supplement 3—source data 2.** *Figure 2—figure supplement 3B*.

**Figure supplement 3—source data 3.** *Figure 2—figure supplement 3C*.

**Figure supplement 4.** *art1*$^{K486R}$ or *art1*$^{ΔPY}$ mutant fused with ubitquitin (Ub) variants do not rescues the cargo protein Can1 sorting defects.

**Figure supplement 4—source data 1.** *Figure 2—figure supplement 4A, B*.

attenuated in *art5*$^{K364R}$ condition (***Figure 2D***). We next asked if the ubiquitination defect of Itr1 is due to the loss of protein-protein interaction between *art5*$^{K364R}$ and Itr1. To test this, Itr1-GFP was co-IPed from yeast strains expressing Art5$^{WT}$ or *art5*$^{K364R}$ (***Figure 2E***). The *art5*$^{K364R}$ can be co-IPed by Itr1-GFP comparable to Art5$^{WT}$, indicating that the decrease of Itrt1 ubiquitination upon inositol stimulation is not due to the loss of interaction between adaptor protein Art5 and cargo protein Itr1.

Consistent with the result of *art5*$^{K364R}$, the *art1*$^{K486R}$ allele leads to a sorting defect of Mup1-GFP (***Figure 2—figure supplement 1A–1C***). We sought to test if the Mup1-GFP can bind to both Art1$^{WT}$ and *art1*$^{K486R}$. To do so, we examined the protein interaction between Mup1 and Art1 using Co-IP analysis. Indeed, we can observe the interaction between Mup1 and overexpressed Art1 (***Figure 2—figure supplement 1D***). In agreement with previous finding that the acidic patch in the Mup1 N-terminal tail is required for binding with Art1 (***Guiney et al., 2016***), we showed that the Q49R Mup1 mutant does not interact with Art1 (***Figure 2—figure supplement 1E***). Furthermore, both Art1$^{WT}$ and *art1*$^{K486R}$ can bind to Mup1, as evidenced by the Co-IP of *art1*$^{K486R}$ with Mup1 when Art1 ubiquitination is impaired (***Figure 2—figure supplement 1F***). Thus, our results demonstrate that the sorting defect of Mup1-GFP in the presence of *art1*$^{K486R}$ is not due to the loss of protein interaction between the adaptor protein Art1 and cargo protein Mup1.

Since TORC1 kinase regulates the Art1-dependent Ub-mediated cargo protein endocytosis by modulating Art1 phosphorylation via Npr1 kinase (***MacGurn et al., 2011***), we next tested if the non-ubiquitinated pool of Art1 altered the Npr1 dependence for phosphorylation, thereby affecting cargo protein sorting. First, we expressed Art1$^{WT}$ or *art1*$^{K486R}$ in WT and *npr1Δ* mutant strains. We observed that both the di-ubiquitinated or the non-ubiquitinated Art1 pools migrated slightly faster in the *npr1Δ* mutant, consistent with dephosphorylation (***Figure 2—figure supplement 2A***). Next, we treated the cells with either rapamycin or cycloheximide to monitor the change in phosphorylation status for ubiquitinated or non-ubiquitinated Art1. Also, the activated Npr1 kinase triggered by rapamycin treatment leads to phosphorylation of both Art1$^{WT}$ and *art1*$^{K486R}$, whereas the dephosphorylation of these two proteins is observed following cycloheximide treatment (***Figure 2—figure supplement 2B***). The Npr1-dependent phosphorylation is therefore the intrinsic feature of Art1, regardless of the ubiquitination status of Art1.

Since ARTs ubiquitination is necessary for cargo sorting, we next sought to test if C-terminal Ub fusion with KR or ΔPY motif mutants of adaptors rescues their cargo sorting defect. We first tested if Itr1-GFP sorting can be restored by *art5*$^{K364R}$ and *art5*$^{ΔPY}$ with C-terminal 2xUb and found the 2xUb fusion does not restore the Itr1 sorting defect (***Figure 2—figure supplement 3A–3D***). Since the toxic arginine analog canavanine hypersensitivity occurs in an *art1Δ* mutant, where Can1 cannot be endocytosed, it provides a readout of Art1 function (***Grenson et al., 1966***). We found that the C-terminal Ub fusions to *art1*$^{K486R}$ or *art1*$^{ΔPY}$ did not enhance their function (***Figure 2—figure supplement 4A–4B***).

These results indicate that di-Ub needs to be conjugated at specific residues for proper functionality. Furthermore, the ubiquitination of ARTs is required for enhancing cargo protein sorting but cannot bypass PY motif, responsible for binding with Rsp5.

## PM recruitment of Rsp5 is enhanced by Art5 and Art1 protein ubiquitination

The $art5^{K364R}$ mutant partially blocks the ubiquitination and cargo sorting of Itr1 after inositol treatment, but still interacts with Itr1. We therefore hypothesized that the defective ubiquitination of $art5^{K364R}$ may impair Rsp5 recruitment to the PM. To test this idea, we examined the localization of Art5-GFP in yeast cells before and after inositol treatment. The Art5$^{WT}$-GFP localized at cytosol, nucleus, and occasional cytosolic puncta (Sec7-negative, *Figure 3A*). Strikingly, the Art5$^{WT}$-GFP is re-localized to PM puncta and patch structures after 30 min of inositol (20 µg/mL) treatment. In comparison to Art5$^{WT}$, the $art5^{K364R}$-GFP and $art5^{\Delta PY}$ mainly remain in the cytosol even after inositol treatment (*Figure 3B-C*). We next asked if Rsp5 can be re-localized to the PM in an Art5-dependent manner after adding inositol to the growth media. As expected, Rsp5 was observed to be recruited to PM patches after inositol treatment in WT cells. However, Rsp5 PM recruitment after inositol treatment is reduced in cells expressing either $art5^{K364R}$ or $art5^{\Delta PY}$ (*Figure 3D-E*).

In parallel to Art5, we also examined the PM localization of Art1 upon methionine treatment. We found that Art1 is efficiently recruited to the PM in rich medium yeast extract peptone dextrose (YPD) or in minimal medium containing methionine (*Figure 3—figure supplement 1A*). In contrast to Art1$^{WT}$, the recruitment of $art1^{K486R}$ to the PM is attenuated and no PM recruitment is seen with $art1^{\Delta PY}$ (*Figure 3—figure supplement 1B–1D*). We next tested whether Art1 facilitates PM recruitment of Rsp5. As expected, methionine treatment induces Rsp5 PM recruitment in cells expressing Art1$^{WT}$, but this recruitment is much reduced in cells expressing $art1^{K486R}$ (*Figure 3—figure supplement 1E–1F*). In agreement with this data, the Rsp5 PM recruitment is more significant when overexpressing Art1$^{WT}$ than $art1^{K486R}$ or $art1^{\Delta PY}$ (*Figure 3—figure supplement 1G–1H*). Taken together, our results support the model that specific ubiquitination of adaptor proteins is required for proper recruitment of Rsp5 to target membranes and subsequent Ub-mediated endocytosis of cargo proteins.

## Substrate dependent PM recruitment of adaptor protein Art5 and Art1

We next sought to examine if cargo proteins are required for adaptor protein recruitment to their functional locations. To test it, we examined the Art5 localization in *ITR1-WT* and *itr1Δ* mutants upon inositol treatment. Strikingly, we found that the PM recruitment of Art5-GFP is abolished in the *itr1Δ* mutant (*Figure 3F and G*). Similarly, we observed that the PM recruitment of Art1 is attenuated in the *mup1Δ* mutant with methionine induction (*Figure 3—figure supplement 2A–2B*). We further tested whether Art1 can be recruited to the PM in cells expressing *mup1*-Q49R mutant upon methionine treatment. Previous data showed that Mup1 mutant Q49R is unable to be endocytosed with methionine treatment (*Guiney et al., 2016*) and the *mup1*-Q49R mutation abolishes the protein-protein interaction between Mup1 and Art1 (*Figure 2—figure supplement 1E*). We observed that PM recruitment of Art1 is abrogated in the *mup1*-Q49R condition (*Figure 3—figure supplement 2A–2B*), suggesting that the Mup1-Art1 interaction is required for methionine-induced Art1 PM re-localization.

## Rsp5 exosite is required for binding between K63-linked di-Ub and HECT domain

We hypothesized that adaptor di-ubiquitination enhances protein-protein interactions between di-ubiquitinated adaptors and Rsp5 and thus promoting the recruitment of the E3 ligase. To test this hypothesis, we set out to examine the binding between mono-Ub or K63-linked di-Ub and the HECT domain of Rsp5. To do so, we first generated K63-linked Ub chains using K63-chain specific E2 enzymes Mms2/Ubc13 (*Hofmann and Pickart, 1999*; *Sato et al., 2008*; *Spence et al., 1995*). We then performed a binding assay between glutathione-*S*-transferase (GST) fusion proteins to Rsp5 HECT domain or GST only and the K63-linked Ub chains. The mono-Ub and K63-linked di-Ub chains bind to GST-HECT domain (lane 7), but not to GST (*Figure 4A*). The binding between mono-Ub and HECT domain depends on the exosite/Ub interface (Y516 and F618) (*French et al., 2009*; *Kim et al., 2011*). We found that the binding between K63-linked di-Ub and HECT domain is disrupted

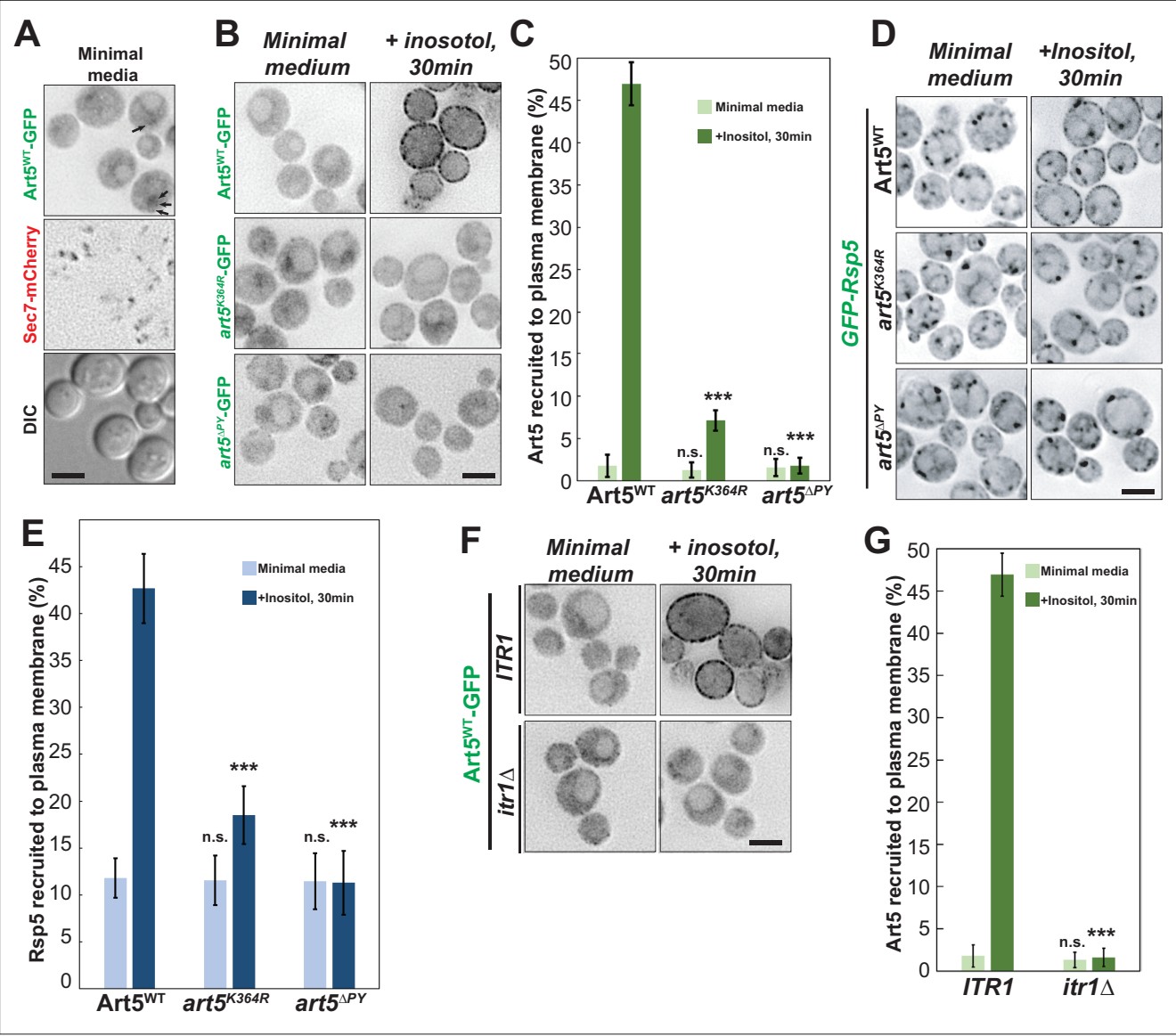

**Figure 3.** Rsp5 plasma membrane (PM) recruitment is enhanced by Art5 ubiquitination. (**A**) Fluorescent microscopy of Art5-GFP with Sec7-mCherry in the WT cells. Black arrows represent occasional cytosolic Art5-GFP dots. (**B**) Fluorescence microscopy of cells expressing Art5$^{WT}$, art5$^{K364R}$, and art5$^{\Delta PY}$ with C-terminal GFP fusion proteins in minimal media and after inositol treatment (20 µg/mL) for 30 min. (**C**) Quantification of PM localization of the indicated Art5$^{WT}$, art5$^{K364R}$, and art5$^{\Delta PY}$ mutants in (**B**). (**D**) Localization of GFP-Rsp5 in the presence of Art5$^{WT}$, art5$^{K364R}$, and art5$^{\Delta PY}$ mutants before and after inositol treatment (20 µg/mL) for 30 min. (**E**) Quantification of the PM localized Rsp5 in the Art5$^{WT}$, art5$^{K364R}$, and art5$^{\Delta PY}$ conditions in (**D**). (**F–G**) Fluorescence microscopy and quantification analysis of the Art5-GFP recruited to PM (%) in the *ITR1* and *itr1Δ* mutant condition. Error bars indicate 95% CI, n=40 cells. ***, p<0.005; n.s., not significant vs Art5 PM recruitment in Art5$^{WT}$ condition in (**C**), vs Rsp5 PM recruitment in Art5$^{WT}$ condition in (**E**), or vs Art5 PM recruitment of *ITR1* condition in (**F**). Scale bar = 2 µm.

The online version of this article includes the following source data and figure supplement(s) for figure 3:

**Source data 1.** *Figure 3C*.

**Source data 2.** *Figure 3E*.

**Source data 3.** *Figure 3G*.

**Figure supplement 1.** The Art1 di-ubiquitination facilitates Rsp5 plasma membrane (PM) recruitment upon methionine treatment.

**Figure supplement 1—source data 1.** *Figure 3—figure supplement 1D*.

**Figure supplement 1—source data 2.** *Figure 3—figure supplement 1F*.

**Figure supplement 1—source data 3.** *Figure 3—figure supplement 1H*.

**Figure supplement 2.** Substrate dependent plasma membrane (PM) recruitment of adaptor protein Art1.

*Figure 3 continued on next page*

*Figure 3 continued*

**Figure supplement 2—source data 1.** *Figure 3—figure supplement 2B*.

by the exosite mutants Y516A, F618A, or the Y516A/F618A double mutant (lanes 8–10, *Figure 4A*), suggesting that the K63-linked di-Ub also interacts with the HECT domain via the exosite.

Since we have shown that adaptor proteins are di-ubiquitinated in a K63-linkage, we next decided to examine the binding affinity between HECT domains and mono-Ub and K63-linked di-Ub. The dissociation constant (Kd) for the interaction between HECT and mono-Ub was quantified by isothermal titration calorimetry (ITC) assay to be approximately 201 µM (*Figure 4B*). Ub is often recognized through a hydrophobic surface containing Ile44 (*Dikic et al., 2009*; *Shih et al., 2000*; *Sloper-Mould et al., 2001*). As expected, the I44A mutation of Ub abolishes the binding between mono-Ub and the HECT domain (*Figure 4—figure supplement 1A*). Our results suggest that the HECT domain exosite and the I44-containing Ub hydrophobic surface are required to bridge the protein-protein interaction between the HECT domain and ubiquitin. In contrast to the mono-Ub results, K63-linked di-Ub enhances the HECT domain binding affinity (Kd = 33 µM), nearly sixfold relative to the mono-Ub (*Figure 4C*). Head-to-tail M1-linked di-Ub was proposed to mimic the K63 Ub linkage (*Komander et al., 2009*; *Zhu et al., 2017*). As expected, our ITC analysis showed that M1-linked di-Ub binds to HECT with Kd = 36 µM (*Figure 4D*), comparable with the K63-linked di-Ub. Also, our in-vitro binding assay showed that the binding between GST-2xUb and HECT domain is stronger than GST-Ub (*Figure 4E*). In comparison, K48-linked di-Ub shows a much lower affinity than K63-di-Ub, Kd = 145 µM (*Figure 4F*). In line with the GST-binding result (*Figure 4A*), no binding was detected between the Ub variants and exosite mutant F618A (*Figure 4—figure supplement 1B-E*). Thus, our results demonstrate that HECT domain specifically binds to linear form K63-linked di-Ub and the exosite site is required for Ub binding.

We next wondered if both the proximal and distal end Ub of the K63-linked di-Ub contribute the binding to the HECT domain. To test it, we fused a distal end Ub (I44A) mutant to a proximal Ub (WT) and generated the distal end I44A mutant of K63 di-Ub (Ub$^{WT}$-Ub$^{I44A}$, proximal-distal). However, since the Ile44 residue of the proximal end Ub is essential for Ub binding by Ubc13/Mms2 and critical for K63-linked di-Ub catalysis, the Ile44 mutant of the proximal end Ub of the K63 di-Ub cannot be made (*Tsui et al., 2005*). We found that the K63-linked Ub$^{WT}$-Ub$^{I44A}$ binds to HECT with a Kd = 120 µM, lower binding affinity than the K63 di-Ub (*Figure 4G*, *Figure 4—figure supplement 1F*). Together, our result suggests that both distal and proximal ubiquitins contribute to the HECT domain binding.

## K63-linked di-ubiquitination enhances the interaction between adaptor proteins and Rsp5

We next sought to determine if K63-linked di-Ub enhances the binding between adaptor and HECT type E3 ligase. We first confirmed the interaction between *art1*-K486R or *art5*-K364R with Rsp5 using Co-IP, due to the interaction between the PY motifs and WW domains (*Figure 5A-B*). Indeed, Art1 and Art5 PY motif containing peptides interact with purified WW domains from Rsp5 (Kd = 3.6 µM for Art1, Kd = 3.1 µM for Art5), but not with PY motif mutants (*Figure 5C-D*, *Figure 5—figure supplement 1A–1B*). We next test if the di-Ub/HECT interaction enhances the binding between E3 ligases and adaptors. We found that we could not express Art1 or Art5 at high levels in *Escherichia coli*, therefore we expressed the Art1 ortholog Any1 from *Schizosaccharomyces pombe* in *E. coli*. The *S. pombe* Rsp5 ortholog Pub1 interacts with Any1 with a binding affinity Kd~2.1 µM (*Figure 5E*), in a similar range as the binding affinity between PY motifs and WW domains shown earlier (*Figure 5C-D*). The association between Pub1 and Any1 was also demonstrated by size exclusion chromatography, in which the Any1 can be co-fractionated with Pub1 without being conjugated with di-Ub (*Figure 5—figure supplement 2A–2C*). The ITC and co-fractionation results suggest that Pub1 interacts with Any1 in a PY motif dependent manner. The di-Ub modification of adaptor protein likely upregulates the interaction affinity between the adaptor and Need4 E3 ligase. Remarkably, Any1 conjugated with K63 di-Ub enhances the binding with Pub1 nearly eightfold in comparison with non-conjugated Any1 (*Figure 5F*), suggesting that di-Ub conjugation onto Any1 probably leads to a structural conformation change of Any1 and therefore enhances the Pub1 binding. This data is in line with the results that di-ubiquitination of Art5 and Art1 are required for efficient Rsp5 recruitment to the PM and for cargo protein sorting. In comparison, the Any1-di-Ub binds to exosite mutant *pub1*-F576A with a Kd =

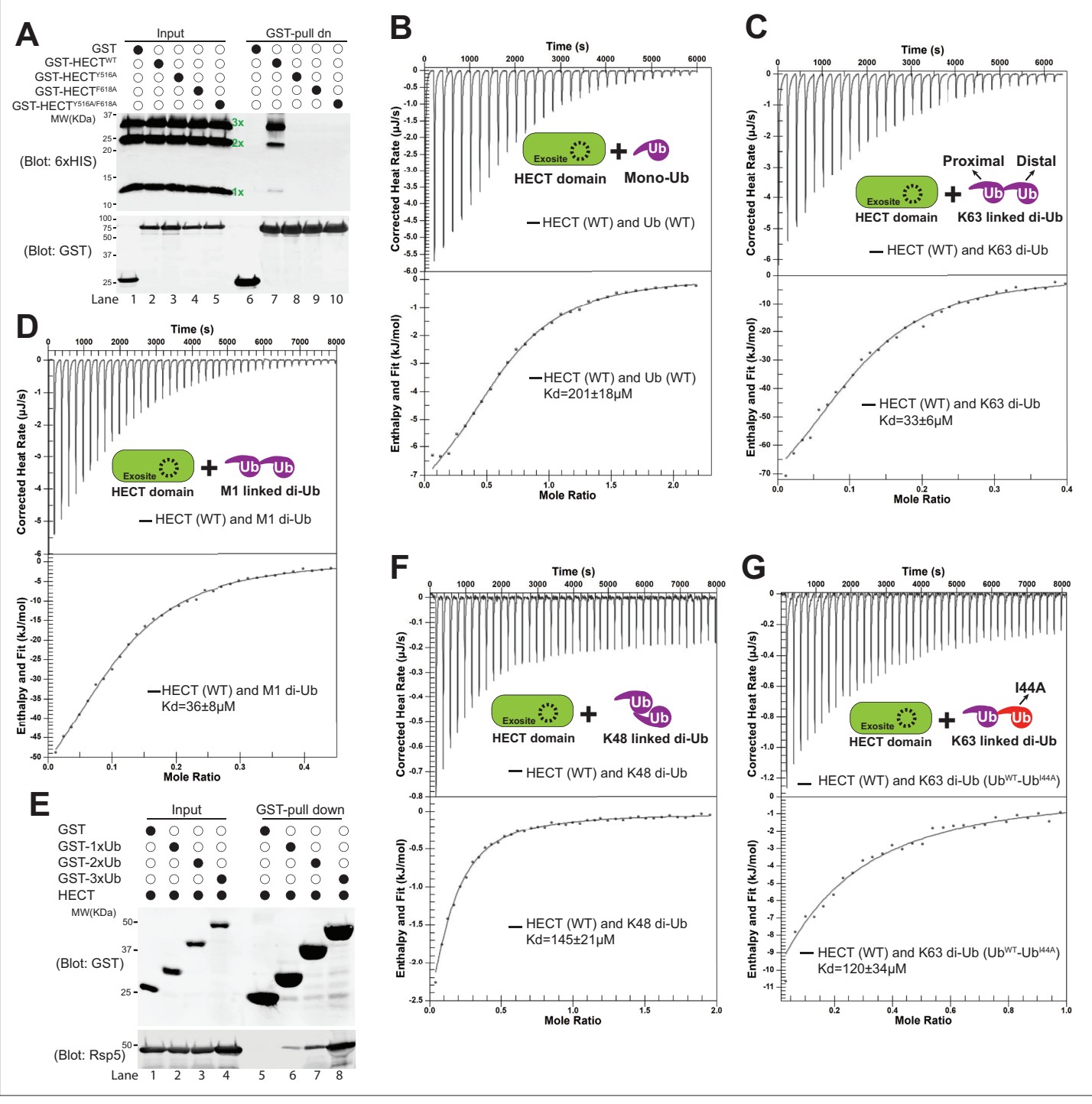

**Figure 4.** K63-linked di-ubiquitin (di-Ub) binds with Rsp5 HECT domain. Rsp5 exosite is required for K63-linked di-Ub binding with HECT domain. (**A**) Glutathione-*S*-transferase (GST) pull down assay between HECT-wild-type (HECT-WT), Y516A, F618A, or Y516A/F618A mutant and K63-linked Ub ladder. (**B**) Example isothermal titration calorimetry (ITC) titration curves showing the binding of mono-Ub-WT or I44A mutant to Rsp5 HECT domain. (**C**) ITC-based measurements of the bindings between K63 di-Ub and Rsp5 HECT domain. (**D**) The representative ITC curves of showing the binding of M1-linked di-Ub and Rsp5 HECT domain. (**E**) GST pull down assay between GST only, GST-1xUb, 2xUb, or 3xUb, and Rsp5 HECT domains. (**F**) Measurement of affinity between K48 di-Ub and Rsp5 HECT domain by ITC. (**G**) ITC-based measurements showing that the K63 di-Ub with a distal end ubiquitin mutant (I44A) partially disrupts the binding affinity with Rsp5 HECT domain.

The online version of this article includes the following source data and figure supplement(s) for figure 4:

**Source data 1.** *Figure 4A and E*.

**Figure supplement 1.** Rsp5 exosite is required for K63-linked di-ubiquitin (di-Ub) to bind with HECT domain.

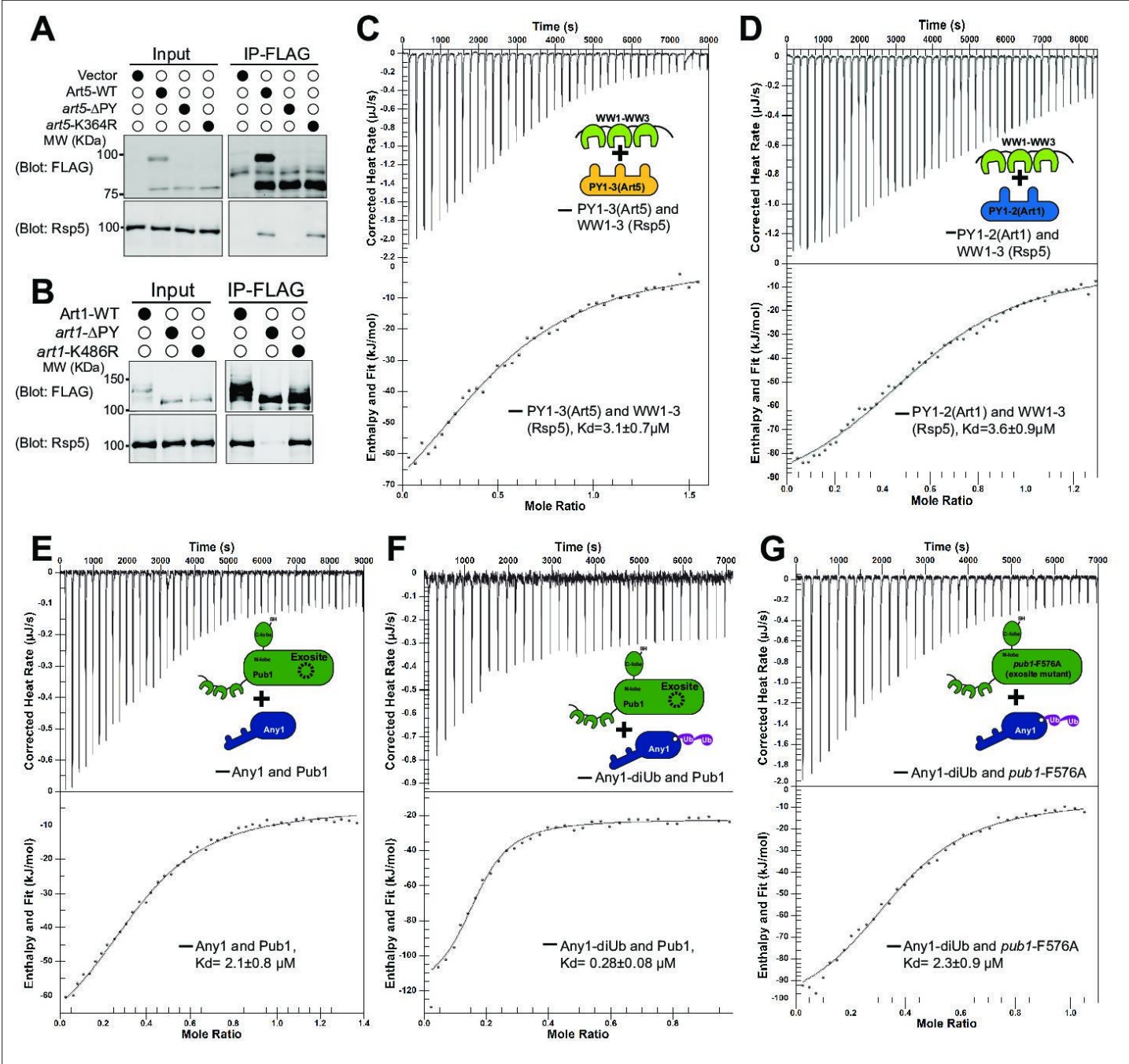

**Figure 5.** K63-linked di-ubiquitination enhances the interaction between adaptor proteins and Rsp5. (**A–B**) Co-IP of Art1 and Art5, WT, KR, and PY motif mutants with Rsp5. (**C–D**) Isothermal titration calorimetry (ITC) analysis of Art1 or Art5 PY motifs containing domain and Rsp5 WW1-HECT domain. (**E**) Analysis of binding affinity between Any1 (Art1 ortholog in *Schizosaccharomyces pombe*) and the Pub1 (Rsp5 orthologue in *S. pombe*). (**F–G**) ITC results obtained by titration of Any1 conjugated with K63 di-ubiquitin (di-Ub) into WT or exosite mutant F576A of Pub1.

The online version of this article includes the following source data and figure supplement(s) for figure 5:

**Source data 1.** *Figure 5A and B*.

**Figure supplement 1.** Isothermal titration calorimetry (ITC) analysis between Rsp5 tryptophan-tryptophan (WW) domains and PY motif mutants of Art1 and Art5.

**Figure supplement 2.** Size exclusion chromatography analysis of the Any1-Pub1 complex.

**Figure supplement 2—source data 1.** *Figure 5—figure supplement 2A, B, C*.

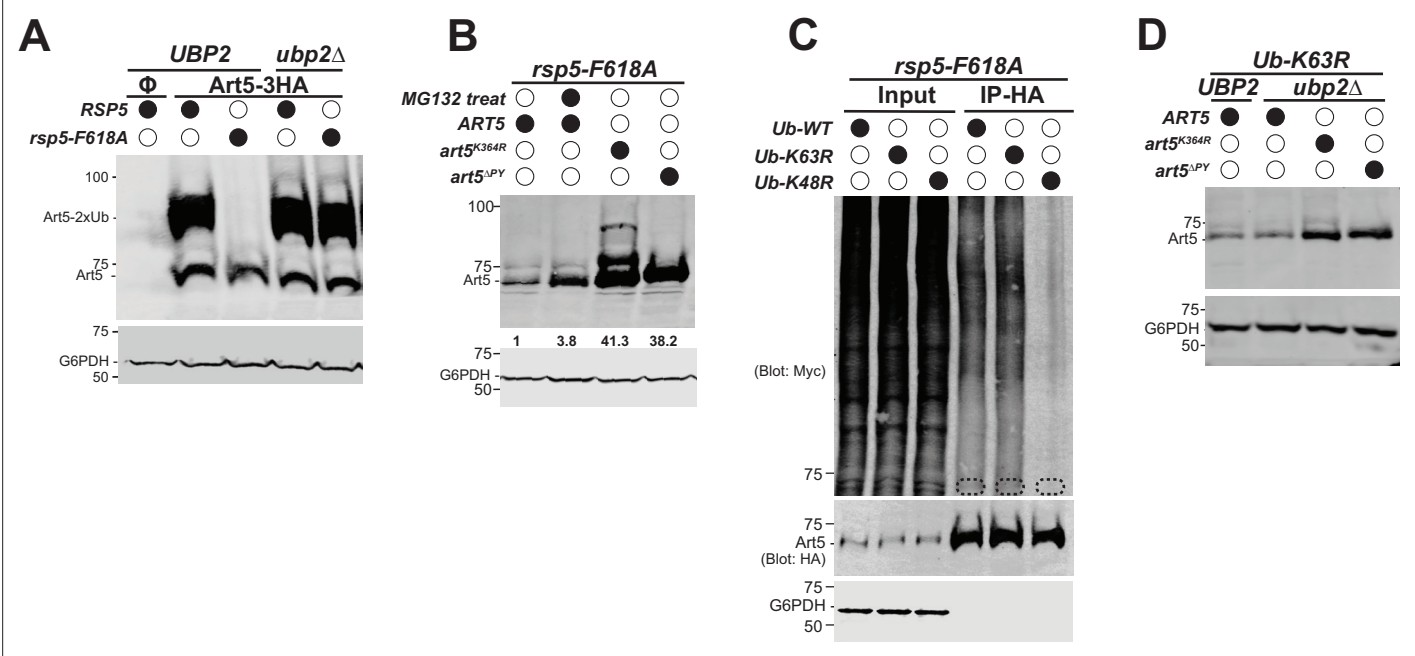

**Figure 6.** Deubiquitination of Art5 di-ubiquitin (di-Ub) by Ubp2. (**A**) Immunoblot analysis of Art5-3HA in the indicated yeast strains: *RSP5/UBP2*, *rsp5-F618A/UBP2*, *RSP5/ubp2Δ*, and *rsp5-F618A/ubp2Δ*. *RSP5/UBP2* cells bearing empty vector is used as a negative control in the first lane. The lanes with Φ symbol indicate negative controls lack of Art5 expression vectors. (**B**) Immunoblot analysis of Art5-3HA in *rsp5*-F618A mutant treated with DMSO or MG132 (25 μg/mL) for 60 min. *art5*[K364R]-3HA and *art5*[ΔPY]-3HA are shown as controls. (**C**) Ub blot of *rsp5*-F618A yeast cells carrying Art5-3HA, as well as WT, K63R, or K48R Myc-ubiquitin expression vector. Cells were treated with MG132 (25 μg/mL) for 2 hr. Samples were immunoprecipitated (IPed) using anti-HA resin and analyzed by immunoblot. The dashed circles highlight the positions of non-ubiquitinated Art5-3HA. (**D**) Immunoblot analysis of Art5-3HA, *art5*[K364R]-3HA, and *art5*[ΔPY]-3HA expressed in *Ub-K63R* and *Ub-K63R/ubp2Δ*.

The online version of this article includes the following source data and figure supplement(s) for figure 6:

**Source data 1.** *Figure 6A, B, C and D*.

**Figure supplement 1.** Deubiquitination (DUB) of K63 di-ubiquitin (di-Ub) of adaptor protein Art1 by Ubp2.

**Figure supplement 1—source data 1.** *Figure 6—figure supplement 1A, B, C, D, E*.

**Figure supplement 2.** The Ubp2-mediated deubiquitination (DUB) recycle of the ubiquitinated Art1 and Art5.

**Figure supplement 2—source data 1.** *Figure 6—figure supplement 2A and C*.

**Figure supplement 2—source data 2.** *Figure 6—figure supplement 2B*.

**Figure supplement 2—source data 3.** *Figure 6—figure supplement 2D*.

2.3 μM (*Figure 5G*), confirming that the di-Ub binds to Pub1 exosite. Together, the di-ubiquitination of adaptor proteins enhances the binding affinity with the E3 ligase, leading to E3 ligase recruitment and cargo protein ubiquitination and sorting.

## DUB of K63 di-Ub of adaptor protein Art5 by Ubp2

Since Rsp5 exosite is essential for binding with the K63-linked di-Ub, we next examined the ubiquitination status for the adaptor proteins Art1 and Art5 in exosite mutants. The di-ubiquitinated form of Art5 is diminished in the *rsp5*-F618A mutant (*Figure 6A*, lane 3). Similarly, the di-ubiquitinated pool of Art1 is substantially attenuated in either the Y516A or F618A exosite mutant (*Figure 6—figure supplement 1A*). It was reported that exosite mutants do not alter the binding affinity between E3 and E2 enzymes, the transthiolation process from E2 to E3, or the self-ubiquitination activity of Nedd4 (*Maspero et al., 2011*). We speculated that a DUB enzyme is involved in the trimming process of the K63-linked di-Ub. To test it, we performed a multicopy gene suppression screen with yeast DUBs and found that overexpressing *UBP2* leads to a reduction of the di-ubiquitinated Art1 (*Figure 6—figure supplement 1B*). Furthermore, the di-Ub modification of Art1 is restored when the catalytic dead mutant C745V of Ubp2 is overexpressed (*Figure 6—figure supplement 1C*). This result infers that Ubp2 may function as a DUB to trim the di-ubiquitinated form of Rsp5 adaptor proteins.

To investigate the role of Ubp2 in the modification of Rsp5 adaptor proteins, we examined Art5 in a double mutant of *rsp5*-F618A/*ubp2*Δ. Strikingly, the di-ubiquitinated Art5 and Art1 are nearly fully restored in the *rsp5*-F618A/*ubp2*Δ strain (*Figure 6A*, *Figure 6—figure supplement 1D*), indicating that Ubp2 trims the di-Ub on adaptors Art5 and Art1 when they are disengaged from the Rsp5 exosite. To test if Ubp2 is playing a catalytic or structural role in this process, we complemented the *rsp5*-F618A/*ubp2*Δ with either a WT or a catalytic mutant *ubp2*-C745V. We found that the Ubp2-WT abrogates the rescue of Art1 trimming seen in the lane 1, whereas the *ubp2*-C745V does not (*Figure 6—figure supplement 1E*). Together, these results suggest that the exosite can protect the di-Ub moiety on adaptors from the cleavage by Ubp2.

We next wondered if the loss of Art5 (*Figure 6A*, lane 2) is mediated by proteasome function. To test it, we treated the *rsp5*-F618A mutant with proteasome inhibitor MG132. We found that the full length Art5 protein is restored 3.8-fold upon the inhibition of proteasome function by MG132 (*Figure 6B*), suggesting that Art5 probably undergoes K48-linked polyubiquitination because K48-linked Ub chains are preferred by proteasome. Either the PY motif or the K364R mutant rescues the loss of Art5 (*Figure 6B*), indicating that Rsp5 is responsible for Art5 degradation and the K364 is used for this ubiquitination process. To directly determine the involvement of K63 versus K48 linkage in the Art5 degradation, we examined the effect of overexpressing myc-Ub with WT, K63R, and K48R mutations on Art5 ubiquitination. We found that expressing myc-Ub K63R does not affect Art5 hyperubiquitination in the *rsp5*-exosite mutant background, whereas the K48R Ub mutant substantially reduced Art5 ubiquitination (*Figure 6C*). This result suggests that the Art5 ubiquitination in the *rsp5*-exosite mutant is mediated by a K48-linked polyubiquitin chain.

We then sought to uncover the mechanism by which the Art5 degradation is triggered. We observed that Art5 protein is also degraded in the *Ub*-K63R mutant (*Figure 1C*). We wondered if *ubp2*Δ can rescue Art5 degradation in the *Ub*-K63R mutant. To test this, we deleted Ubp2 in the *Ub*-K63R mutant and found that *ubp2*Δ does not reverse the loss of Art5 protein (*Figure 6D*). Our results suggest that the loss of K63-linked di-Ub on Art5, instead of Ubp2, in either the *Ub*-K63R mutant or *rsp5*-exosite mutant, leads to Art5 degradation. Also, in these two conditions, the Art5 degradation can be rescued in *art5*-K364R and *art5*-ΔPY mutant (*Figure 6B and D*).

We next asked if the adaptor ubiquitination process is reversible and Ubp2 is involved in this process or not. To monitor the pre-existing adaptor proteins, we decided to employ the *tet*-Off system to fix the pool of adaptor proteins by treating the cells with doxycycline. As seen in the *Figure 6—figure supplement 2A–2D*, pre-existing Art5 or Art1 undergoes ubiquitination upon inositol or methionine treatment for 1 hr in both WT and *ubp2*Δ conditions, whereas the adaptor proteins shifted back to less ubiquitinated status after removing the stimulation in the WT condition, but not in the *ubp2*Δ mutant. Together, our data demonstrate a model for adaptor protein recycling mediated by Ubp2 (*Figure 6—figure supplement 2E*). First, stimulation enhances adaptor ubiquitination. Second, the ubiquitinated pool of adaptor proteins can be deubiquitinated by Ubp2 when stimulation is terminated.

## Discussion

In this study, we identified the first K63-linked di-Ub modification that modulates the function of Rsp5 and adaptor proteins. Our data demonstrates that two biological functions are implicated with this K63-linked di-Ub modification. First, K63-linked di-Ub activates Rsp5 function. K63-linked di-Ub enables the full engagement of adaptors onto the Rsp5 exosite and sharply enhances the binding affinity with Rsp5, which facilitates Rsp5 recruitment and accelerates substrate protein ubiquitination. Second, K63-linked di-Ub on adaptors engaged with the Rsp5 exosite are not accessible to Ubp2. Once released from Rsp5 exosite, the exposed K63-linked di-Ub is subjected to cleavage by Ubp2. Furthermore, we monitored the ubiquitination status of adaptor proteins Art1 and Art5. Using *tet*-Off system, we have shown that adaptor proteins undergo ubiquitination upon substrate stimulation and Ubp2 is required for DUB of adaptor proteins once the stimulation is removed. As hypothesized by our earlier review (*MacGurn et al., 2012*), our current data supports the model that ubiquitinated adaptor proteins were deubiquitinated by Ubp2 so that the adaptor proteins can be recycled for the next round of ubiquitination event.

## K63-linked di-Ub is engaged into Rsp5 E3 ligase for activation

While we showed that Rsp5 adaptors Art1, Art4, and Art5 undergo K63-linked di-Ub modification, we also demonstrate that this conjugation sharply enhances the binding with the E3 ligase and activates the E3 ligase function for substrate ubiquitination. We reason that the interaction between the di-Ub chain and the HECT domain locks the E3 ligase and adaptor into an active/functional conformation. For adaptor-independent ubiquitination, the Nedd4/Rsp5 ligase exosite is also required for efficient Ub conjugation, demonstrating that the 'Ub-exosite binding' is required to localize and orient the distal-end Ub chain to promote conjugation (*Kim et al., 2011*; *Maspero et al., 2011*). In terms of the Rsp5 adaptor-mediated function, we propose that the binding between 'di-Ub and exosite' not only enhances the binding affinity between the E3 ligase and adaptor (*Figure 5E, F*), but also leads to more productive Rsp5 recruitment to properly orient and present the substrate for ubiquitination at target membranes.

While we presented the evidence of E3 ligase activation by ubiquitinated adaptors, we also showed that K63 di-Ub generates a sixfold tighter binding to the HECT domain than mono-Ub. We reason that the K63 di-Ub not only provides alternative options to bind a single site, but also fits with a model in which there are multiple Ub binding sites. It was found that three N-lobe mutations (Y516A, F618A, and V621A/V622A) completely abolished Ub binding and three extra mutations (N513A, Y521A, and R651A) caused a reduction in binding (*French et al., 2009*). Kim and coworkers found that the L8-I44-V70 hydrophobic patch of mono-Ub sits on Rsp5 in three legs, like a tripod (*Kim et al., 2011*). Likewise, two separated UIMs in Rap80 bind to extended K63-linked Ub chain favorably (*Sato et al., 2009*; *Sims and Cohen, 2009*). Indeed, K63-linked di-Ub with a mutation (I44A) at the distal end Ub leads to lower binding with Rsp5 (*Figure 4G*). We propose that Rsp5 exosite accommodates the two hydrophobic patches of the distal and proximal Ub at multiple Ub binding sites, which needs be assigned in the future by structural analysis.

## The linkage specificity and length control for the K63-linked di-Ub

We have been intrigued by the question of how the K63 linkage of di-Ub was achieved and preferred, instead of K48. While the K63-linked di-Ub binds to the HECT domain with a stronger affinity than K48-linked di-Ub (*Figure 4*), short K63 di-Ub chains (<4) generally do not play as degradation signal (*Nathan et al., 2013*; *Thrower et al., 2000*; *Windecker and Ulrich, 2008*). Interestingly, both the M1- and K63-linked di-Ub adopt an equivalent open conformation (*Komander et al., 2009*) and exhibit similar binding affinity to the HECT domain (*Figure 4*), indicating that the HECT domain exosite has a strong preference for the linear and extended form of di-Ub. In contrast, the K48-linked polyubiquitin chain adopts a significantly distinct and compact structure (*Eddins et al., 2007*), which may not be favorable for the HECT domain.

Why is the K63-linked di-Ub chain limited to a dimer? On the one hand, this probably correlates with the physiological reversible function of adaptors. The K63-linked Ub chains (≥3) likely generate stronger binding with the HECT domain than di-Ub. We reason that the di-Ub binds well with HECT domain, but still can be disengaged from the HECT domain under physiological conditions so that Rsp5 can be disassociated and recycled. Furthermore, the K63-linked di-Ub is probably just enough to be masked by the HECT domain exosite cavity whereas longer chains will be trimmed by Ubp2. Future studies could address the accessible region for the di-Ub isopeptide bond cleavage by Ubp2 when di-Ub is engaged into the HECT domain.

## Ubp2 mediates the recycling of Rsp5 E3 ligases from adaptors after ubiquitination

The PY motif containing Rsp5 adaptors share the E3 ligase Rsp5 and an adaptor should disassociate from Rsp5 to allow other adaptors to engage with Rsp5 to ubiquitinate different substrate proteins. In agreement with this working model, Nedd4-mediated downregulation of the sodium channel ENaC is impaired when Nedd4 is sequestered by overexpression of another Nedd4 E3 adaptor, Ndfip2 (*Konstas et al., 2002*).

Besides cleavage of K63 di-Ub in the *rsp5*-exosite mutant, Ubp2 allows the recycling of Rsp5 from its adaptor proteins. MacDonald and coworkers proposed that several Rsp5 adaptors compete for Rsp5 and a Ubp2 deficiency increased both the adaptor activity and the ability to compete for Rsp5 (*MacDonald et al., 2020*). Indeed, we observe that Art5 and Art1 di-ubiquitination in response to

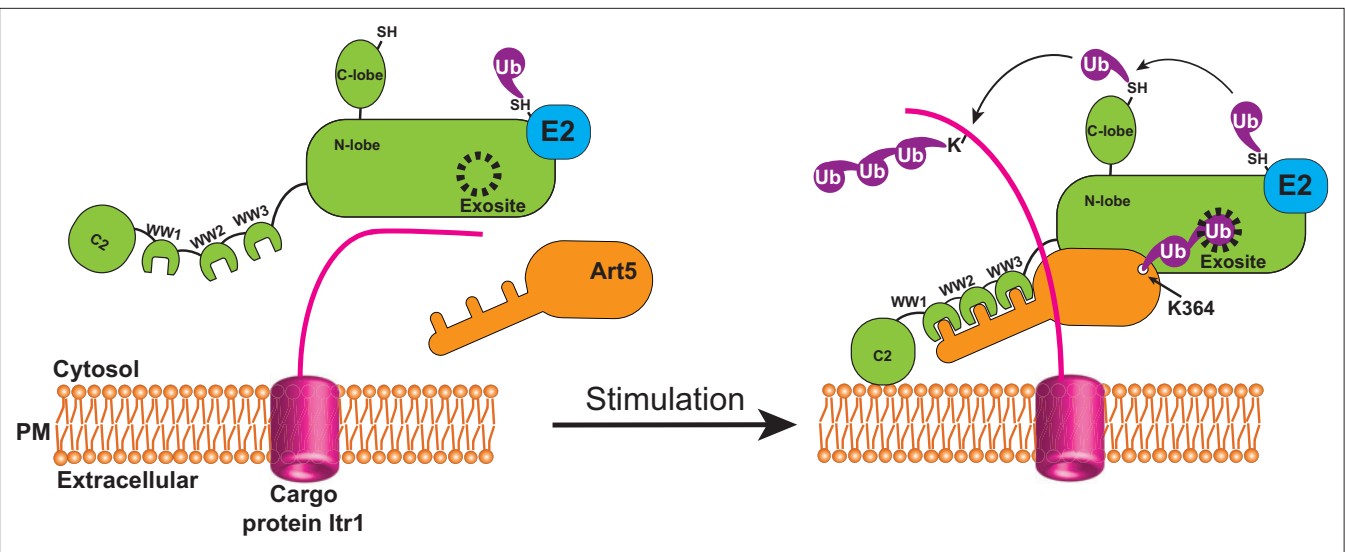

**Figure 7.** Attachment of K63 di-ubiquitin (di-Ub) to adaptor protein Art5 enables efficient membrane recruitment of Rsp5. This model depicts that the adaptor protein Art5 forms binary protein complex with E3 ligase Rsp5 via the interaction between the PY motif and tryptophan-tryptophan (WW) domain, following by the stimulation of cargo protein Itr1. The efficient membrane recruitment of Rsp5 is activated by this binding scaffold when adaptor protein Art5 linked with K63 di-Ub is fully engaged into the Rsp5 exosite.

inositol or methionine treatment is inhibited in the absence of Ubp2 (*Figure 6—figure supplement 2*), indicating the decrease of Rsp5 availability in *ubp2Δ* mutant. Since K63 di-Ub greatly enhances the binding affinity between adaptors and E3 ligase, Ubp2 likely helps the Nedd4/Rsp5 E3 ligase to catalyze distinct ubiquitination events by cleaving the di-Ub off the adaptors and recycling Rsp5. The multitasking of Rsp5 via various adaptors leads us to hypothesize that activated Rsp5 can be released from engaged adaptor proteins. We showed that the adaptor proteins Art1 and Art5 undergo di-ubiquitination upon environmental stimulation and Ubp2 is required to reverse this ubiquitination. Once the ubiquitination is done, the engaged K63 di-Ub is exposed for cleavage by Ubp2. Thereafter, Ubp2 acts on ubiquitinated adaptor proteins to release the adaptor proteins and Rsp5. The mechanism by which Ubp2 executes this reaction needs to be resolved in the future.

Collectively, we propose that Rsp5 ubiquitinates adaptors to trigger their engagement with the Rsp5 exosite upon stimulation, which enables the tight binding between adaptors and Rsp5, and efficient Rsp5 recruitment to target membranes thereby activating Rsp5 function (*Figure 7*). Ubp2 acts as an antagonist for K63 di-Ub to modulate the interaction between K63-di-Ub and the Rsp5 exosite in a reversible manner to maintain cellular homeostasis of Rsp5. Future work needs to address the atomic structure of the ART family of adaptor proteins in complex with Rsp5 to understand how di-Ub is attached to the adaptor and how the di-ubiquitinated adaptors engage with the HECT E3 ligases, stabilizing an activated conformation of the E3 ligase.

## Materials and methods

**Key resources table**

| Reagent type (species) or resource | Designation | Source or reference | Identifiers | Additional information |
|---|---|---|---|---|
| Antibody | Anti-GFP, B2 (mouse monoclonal) | Santa Cruz | sc-9996 | WB (1:2000) |
| Antibody | Anti-Myc (mouse monoclonal) | Santa Cruz | sc-40 | WB (1:2000) |
| Antibody | Anti-GFP (rabbit polyclonal) | Torrey Pines Biolabs | TP401 | WB (1:10,000) |
| Antibody | Anti-HA 12CA5 (mouse monoclonal) | Sigma-Aldrich | 11583816001 | WB (1:5000) |
| Antibody | Anti-FLAG M2 (mouse monoclonal) | Sigma-Aldrich | F1804 | WB (1:5000) |

*Continued on next page*

*Continued*

| Reagent type (species) or resource | Designation | Source or reference | Identifiers | Additional information |
|---|---|---|---|---|
| Antibody | Anti-G6PDH (rabbit, polyclonal) | Sigma-Aldrich | SAB2100871 | WB (1:30,000) |
| Antibody | IRDye® 800CW (Goat anti-Mouse, polyclonal) | LI-COR | 926–32210 | WB (1:10,000) |
| Antibody | IRDye® 800CW (Goat anti-Rabbit, polyclonal) | LI-COR | 926–32211 | WB (1:10,000) |
| Antibody | IRDye® 680LT (Goat anti-Rabbit, polyclonal) | LI-COR | 926–68021 | WB (1:10,000) |
| Antibody | IRDye® 680LT (Goat anti-Mouse, polyclonal) | LI-COR | 925–68070 | WB (1:10,000) |
| Cell line (*E. coli*) | Competent cells of DH5α | ThermoFisher | 18258012 | Super competent cells. |
| Cell line (*E. coli*) | Competent cells of BL21, rosetta | Sigma-Aldrich | CMC0016 | Super competent cells. |
| Software, algorithm | ImageJ | NIH | Version: 1.53 n | https://imagej.nih.gov/ij/ |
| Software, algorithm | NanoAnalyze | TA Instruments | Version: 3.12.0 | https://www.tainstruments.com |
| Software, algorithm | SnapGene | GSL Biotech | Version: 6.0.2 | https://www.snapgene.com |
| Other | cOmplete Protease Inhibitor Cocktail | Roche | 11697498001 | Protease Inhibitors for protein purification. |

## Yeast strains, cloning, mutagenesis and cell growth conditions

The *ART1*, *ART4*, *ART5*, *ITR1*, *MUP1*, and *YUH1* genes were cloned from *Saccharomyces cerevisiae* yeast strain SEY6210. Pub1 and Any1 were PCR amplified from *S. pombe* yeast strain PR109 and subcloned into pET28a with an N-terminal 6xHis-SUMO tag. When necessary, the gene deletions and taggings were made using gene replacement technique with longtine-based PCR cassettes (*Longtine et al., 1998*). All yeast strains and plasmids are described in the *Supplementary file 1*. For fluorescent microscopy experiments, cells were grown overnight to mid-log phase (OD600~0.5) in synthetic media at 30°C. For inositol or methionine stimulation experiments, cells were grown in synthetic media to log phase (OD600~0.8) then treated with exogenous inositol and methionine at different concentrations. Ub-WT, Ub-K63R, Ub-K48R, Ub-D77, Mms2, and Ubc13 were PCR amplified from yeast strain SEY6210 genomic DNA and cloned into pET21a, pET28a-6xHIS, and pGEX6p-1, respectively. The 1x, 2x, and 3x and 4x Ub head-to-tail fusions of Art1, Art4, Art5 expression, and pGEX6p-1 vectors were made by Gibson assembly. E1 enzyme Uba1 (human), E2 enzyme UbcH5C, and K48 Ub linkage specific E2 enzyme E2-25K expression vectors are from our lab stock. YUH1 was subcloned into pGEX6p-1 expression vector with an N-terminal GST tag. PY motifs containing regions for Art1 (661-710) and Art5 (520-586) were PCR amplified and cloned into pGEX-6p-1 vectors. Rsp5 HECT domain (444–809) and WW1-HECT domain (224–809) were fused with N-terminal SUMO tag and cloned into pET28a vector. DUB enzymes USP2, OTUB1, YOD1, AMSH, OTULIN, and Cezanne were purchased from LifeSensors.

## Protein purification

All pET21a, pET28a, pGEX6p-1 constructs were transformed into *E. coli* strain Rosetta (DE3) cells. Single colonies were then cultured in Luria-Bertani medium containing either 100 µg/mL ampicillin or 50 µg/mL kanamycin to a density between 0.6 and 0.8 OD600 at 37°C. Cultures were induced with 0.2 mM isopropyl-B-D-thiogalactopyranoside at 18°C for 16 hr. *E. coli* cells were collected by centrifugation at 3500 rpm for 15 min at 4°C. For non-tagged Ub purification, cells were disrupted by sonication in the lysis buffer (50 mM NH4Ac [pH 4.5], 2 mM DTT, 1 mM EDTA, 1 mM PMSF). For 6xHIS-SUMO tagged proteins, cells were sonicated in the lysis buffer (20 mM Tris-HCl [pH 7.5], 150 mM NaCl, 2 mM DTT, 1 mM EDTA, 1 mM PMSF). For GST fusion proteins, cells were disrupted in the lysis buffer (200 mM NaCl, 25 mM Tris-HCl pH 8.0, room temperature, 2 mM EDTA, 2 mM DTT, 1 mM PMSF).

The lysate for Ub (WT, K63R, K48R, I44A, D77, or D77/I44A) was adjusted to pH 4.5 then spun down at 46,000 × g for 45 min at 4°C. The supernatant was heated at 70°C for 5 min then spun down

again with the same condition. The supernatant was loaded onto SP Sepharose Fast Flow resin pre-equilibrated with the same lysis buffer (pH 4.5). The Ub was eluted with 50 mM NH4Ac (pH 4.5) buffer containing 2 mM DTT using a linear gradient of 0–500 mM NaCl. The eluted Ub mutants were fractionated by Superdex 200 column using size-exclusion buffer (20 mM Tris-HCl [pH 7.5], 150 mM NaCl, 2 mM DTT). Each mutant was concentrated to 15 mg/mL and stored at –80°C.

For 6xHIS-SUMO-tagged (HECT, Pub1 [287–767], Any1 [17–361], and WW1-HECT) and GST-tagged proteins (Ubc13, E2-25K, Yuh1, PY motifs of Art1 or Art5, and M1-linked Ub-Ub), the sonicated lysates were centrifuged 46,000 × g for 45 min at 4°C. The supernatant was bound with TALON cobalt resin or Glutathione Sepharose 4 Fast Flow and the resins were digested by SUMO-specific Ulp1 or GST-specific PreScission proteases to release the proteins of interest. The eluted proteins were fractionated by Superdex 200 using size-exclusion buffer (20 mM Tris-HCl [pH 7.5], 150 mM NaCl, 2 mM DTT). Ubc13, E2-25K, and Yuh1 were concentrated to 750 µM with 20% glycerol and the other proteins were concentrated to 1 mM and stored at –80°C.

For 6xHis-tagged Uba1 and Mms2 purification, the *E. coli* cells were sonicated in lysis buffer 20 mM Tris (pH 7.5), 150 mM NaCl, 2 mM DTT, cOmplete protease inhibitor. The cell lysate (per 1 L) was cleared by centrifugation at 46,000 × g, 45 min, 4°C. The supernatant was incubated with cobalt-chelate TALON resin for 30 min before column wash with lysis buffer supplemented with 25 mM imidazole and the protein of interest was eluted with 300 mM imidazole and dialyzed against 50 mM Tris-HCl (pH 7.6) containing 2 mM DTT and 0.1 mM EDTA. The protein is concentrated to 100 µM with 20% of glycerol and stored at –80°C.

For GST-tagged protein (GST-1xUb, GST-2xUb, and GST-3xUb) purification, the sonicated cell lysate was spun down at 46,000 × g, 45 min, 4°C. The supernatant per 1 L of cells was incubate with 2 mL of Glutathione Sepharose 4 Fast Flow resin and washed with 5 column volumes of wash buffer (20 mM Tris-HCl [pH 8.0], 200 mM NaCl, 1 mM DTT). The GST-tagged proteins were eluted by 2 column volumes of elution buffer (100 mM Tris pH 8.5, 20 mM Glutathione) then dialyzed against size-exclusion buffer (20 mM Tris-HCl [pH 7.5], 150 mM NaCl, 2 mM DTT). Each protein was concentrated to 30 mg/mL and stored at –80°C.

For synthesis of K63 or K48 di-Ub proteins, 5 × PBDM buffer was prepared: 250 mM Tris-HCl (pH 8.0 and 7.6), 25 mM MgCl₂, 50 mM creatine phosphate (Sigma P7396), 3 U/mL of inorganic pyrophosphatase (Sigma I1891), and 3 U/mL of creatine phosphokinase (Sigma C3755). K63-linked di-Ub is synthesized by incubating purified human E1 (0.1 µM), yeast E2 (Ubc13 and Mms2, 8 µM of each), two Ub mutants (K63R and D77, 5 mg/mL of each), ATP (2.5 mM), 1 mM DTT and 1 × PBDM buffer (pH 7.6). For K48-linked di-Ub synthesis, purified human E1 (0.1 µM), E2-25K (20 µM), two Ub mutants (K48R and D77, 7.5 mg/mL of each), ATP (2.5 mM), 1 mM DTT and 1 × PBDM buffer (pH 8.0) were mixed. The reaction K63 or K48 di-Ub were incubated at 37°C for overnight then the reaction was chilled on ice to stop the reaction. About 0.2 volume of 2 M ammonium acetate was added to the reaction to decrease the pH to less than 4.0. The mixture was loaded to SP Sepharose Fast Flow. The K63 di-Ub or K48 di-Ub mixtures were loaded onto Superdex 75 size-exclusion column using gel filtration buffer (20 mM Tris-HCl [pH 7.5], 2 mM DTT, 150 mM NaCl) and the fractions of di-Ub were pooled and concentrated.

## Synthesis and purification of Any1-di-Ub

To remove the D77 of the proximal Ub and unlock the carboxyl-terminal Gly-Gly of K63 di-Ub for further conjugation, purified K63-linked di-Ub (30 mg/mL) is exchanged into hydrolysis buffer (50 mM Tris-HCl [pH 7.6], 1 mM EDTA, and 1 mM DTT) and treated with purified Yuh1 (final concentration of 16 µg/mL) for 60 min at 37°C. After cooling down the reaction at room temperature, 4 mM DTT to the mixture is supplemented with DTT to 5 mM (final concentration). The reaction mixture was then applied to a 5 mL Q column equilibrated with Q buffer (50 mM Tris-HCl pH 7.6, 1 mM EDTA, 5 mM DTT). After 2 volumes of wash, the unbound K63 di-Ub (D77 removed) is collected and concentrated. Di-ubiquitination of Any1 was carried out by incubating purified Any1 proteins with human E1 (0.1 µM), human E2 (UbcH5C, 0.3 µM) and Pub1 (0.3 µM), K63 di-Ub (D77 removed, 10 µM), ATP (2.5 mM), 1 mM DTT and 1 × PBDM buffer (pH 7.6) for 30 min at room temperature. The reaction mixture was chilled on ice before loading onto Superdex 200 size-exclusion column using gel-filtration buffer (150 mM NaCl, 20 mM HEPES [pH 7.5]), and fractions of Any1-di-Ub were pooled and concentrated.

## Analytical size exclusion chromatography analysis

Mix the 20 µM of purified Pub1 and Any1 (non-ubiquitinated) protein in a 175 µL of reaction volume. Pub1 and Any1 protein alone will be used as controls. Incubate the protein samples at room temperature for 30 min. Load 150 µL of sample (in a 1 mL of syringe) to onto a Superdex 200 Increase 10/300 GL column. For UV spectrum detection, the protein fractions absorbance was measured at 280 nm. Each fraction is collected and resolved by 10% SDS-PAGE then stained with Coomassie Blue R250 (0.1% [m/V] in 10% acetic acid, 50% methanol and 40% $H_2O$) for 1 hr with rocking at room temperature. The strained gels were then incubated with destaining solution (10% acetic acid, 50% methanol and 40% $H_2O$).

## GST pull down assay

For pull-down experiments, 2 µM of GST fusion proteins were immobilized onto 100 µL of glutathione bead slurry in the 1 mL of pull-down buffer (50 mM Na-HEPES pH 7.5, 150 mM NaCl, 1 mM EDTA, 1 mM EGTA, 10% Glycerol, 1% Triton X-100, 2 mM DTT). 500 ng of Rsp5 HECT protein was added to the mixture and incubated at 4°C for 2 hr. After 4 washes with pull down buffer, specifically bound proteins were eluted by SDS-sample buffer and resolved on SDS-PAGE (11%) and detection was obtained by Coomassie-staining.

## ITC assay

ITC experiments were carried out on an Affinity-ITC calorimeter (TA instruments) at 25°C. Titration buffer contained 20 mM Tris-HCl (pH 7.5), 150 mM NaCl, 1 mM DTT. For a typical experiment, each titration point was performed by injecting a 2 µL aliquot of protein sample (50–1000 µM) into the cell containing 300 µL of another reactant (5–300 µM) at a time interval of 200 s to ensure that the titration peak returned to the baseline. The titration data was analyzed with NanoAnalyze v3.12.0. using an independent binding model. Coomassie blue stained SDS-PAGE gels showing the purity of proteins used in the ITC experiment are shown in *Supplementary file 2*.

## Fluorescence microscopy assay

For fluorescence microscopy, cells expressing GFP or mCherry proteins were visualized using a Delta-Vision Elite system (GE), equipped with a Photometrics CoolSnap HQ2/sCMOS Camera, a 100× objective, and a Standard Filter Set ('FITC' for GFP fusion protein and 'mCherry' for mCherry fusion proteins). Image acquisition and deconvolution were performed using Softworx.

## Whole cell lysate extraction and Western blotting

Whole cell extracts were prepared by incubating 6 ODs of cells in 10% Trichloroacetic acid on ice for 1 hr. Extracts were fully resuspended with ice-cold acetone twice by sonication, then vacuum-dried. Dry pellets were mechanically lysed (3 × 5 min) with 100 µL glass beads and 100 µL Urea-Cracking buffer (50 mM Tris-HCl [pH 7.5], 8 M urea, 2% SDS, 1 mM EDTA). 100 µL protein 2 × sample buffer (150 mM Tris-HCl [pH 6.8], 7 M urea, 10% SDS, 24% glycerol, bromophenol blue) supplemented with 10% 2-mercaptoethanol was added and samples were vortexed for 5 min. The total proteins were then precipitated by 10% Trichloroacetic acid and washed with cold acetone. The cell pellets were then solubilized by 100 µL Urea-Cracking buffer and mixed with 100 µL of 2× sample buffer. The protein samples were resolved on SDS-PAGE gels and then visualized by immunoblots.

The antibodies and dilutions were used in this study: rabbit anti-G6PDH (1:30,000; SAB2100871; Sigma), rabbit anti-GFP (1:10,000; TP401; Torrypines), mouse monoclonal anti-GFP (1:1000; sc-9996; Santa Cruz), mouse monoclonal anti-Myc (1:5000, sc-40, SantaCruz), IRDye800CW Goat anti-Mouse (1:10,000; 926–32210; LI-COR), IRDye800CW goat anti-rabbit (1:10,000; 926–32211; LI-COR), IRDye680LT goat anti-rabbit (1:10,000; 926–68021; LI-COR) and IRDye680LT goat anti-mouse(1:10,000; 925–68070; LI-COR).

## Immunoprecipitation (IP) assay

100 Ods of cells were collected and washed with water at 4°C. To examine the interaction between Art1 and Mup1-GFP, between Art5 and Itr1-GFP, or between ARTs protein and Rsp5. Yeast cells were washed with ice-cold water three times. The cells were lysed in 500 µL of IP buffer (20 mM Tris-HCl, pH 7.5, 0.5 mM EDTA, pH 8.0, 0.5 mM EGTA, 0.5 mM NaF, 150 mM NaCl, 10% glycerol, 1 mM PMSF,

20 mM NEM, and cOmplete Protease Inhibitor). Cell extracts were prepared by glass-bead beating with 0.5 mm zirconia beads for five cycles of 30 s vortexing with 1 min breaks on ice. Membrane proteins were solubilized in 500 µL IP buffer 1% Triton X-100. The lysates were incubated at 4°C for 30 min with rotation then spun at 500 × g for 5 min at 4°C. The supernatant was clarified by centrifugation at 16,000 × g for 10 min. To detect the interaction between ARTs and Mup1 or Itr1-GFP proteins, the cleared lysate was incubated with 50 µL of GFP-nanotrap resin for 2 hr at 4°C. To examine the interaction between Rsp5 and ARTs, the cleared lysate was bound with 50 µL of FLAG-M2 affinity gel (Sigma, A2220) at 4°C for 4 hr. After incubation, the resin was washed five times with 0.1% Triton X-100 in IP buffer and the bound protein was eluted by 50 µL of 2 × sample buffer.

To examine the ubiquitination of Itr1, cells were grown to early log phase in synthetic media. Yeast strain (*doa4Δpep4Δart5Δ*, Itr1-GFP) cells co-expressing Myc-Ub expression vector (*Zhu et al., 2017*) and Art5$^{WT}$ or *art5$^{K364R}$* were grown to mid-log phase in synthetic medium at 30°C. Cells were pretreated with 0.1 µM CuCl2 for 4 hr to induce the Myc-Ub expression prior to inositol (20 µg/mL) treatment. 100Ods of Cells were incubated with 10% TCA buffer and the extracts were washed with cold acetone. Dry pellets were mechanically lysed (3 × 5 min) with 100 µL glass beads and 100 µL Urea-Cracking buffer (50 mM Tris-HCl pH 7.5, 8 M urea, 2% SDS, 1 mM EDTA, 200 mM NEM). The cell lysates were mixed with 1 mL of IP buffer (50 mM HEPES-KOH, pH 6.8, 150 mM KOAc, 2 mM MgOAc, 1 mM CaCl$_2$, 20 mM NEM, and 15% glycerol) with cOmplete protease inhibitor (Sigma-Aldrich, St. Louis, MO). The cell lysates were clarified by spinning at 16,000 × g for 10 min at 4°C. The resulting lysate was then incubated with 50 µL GFP-nanotrap resin for 4 hr at 4°C. The resin was washed 5 times with 0.1% Triton X-100 in IP buffer. Bound protein was eluted by 50 µL of 2 × sample buffer. Whole cell lysate and the IP reaction was resolved on 10% SDS-PAGE gels and the blots were probed with both GFP and Myc antibodies.

To examine the ubiquitination status of Art4, yeast cells expressing FLAG-tagged Art4 variants were grown to mid-log phase and harvested. 30 ODs per each Art4 protein expressing yeasts were lysed with 500 µL of IP buffer. The cell extracts were solubilized in 500 µL IP buffer supplemented with 1% (v/v) Triton X-100. Art4 proteins were bound with FLAG-M2 affinity gel for 4 hr then treated with Lambda Protein Phosphatase (NEB #P0753) for 1 hr at 37°C. The resulting products were resolved on 7% SDS-PAGE gels and subjected to immunoblot against FLAG antibody.

## In vitro DUB assay

The UbiCRest analysis by linkage selective DUBs was performed as described (*Hospenthal et al., 2015*). 200 ODs of cells expressing Art1-HTF (YMB1005) were collected. Art1 protein is IPed using 200 µL of FLAG-M2 affinity gel and the beads were washed five times with 0.1% Triton X-100 in IP buffer without NEM. Following the last wash, the beads were divided into 10 µL aliquots. Each bead aliquot (with IPed Art1 protein) or 4 µg of purified K63-linked di-Ub was resuspended in a 50 µL of DUB dilution buffer (25 mM Tris-HCl [pH 7.5], 150 mM NaCl, 10 mM DTT) and mixed with or without DUBs: 2.5 µM USP2, 3.0 µM OTUB1, 2.5 µM YOD1, 10.0 µM AMSH, 2.5 µM OTULIN, and 1.0 µM Cezanne. Incubate the DUB reaction tubes at 37°C for 2 hr with gentle mixing. After incubation, cleavages were terminated by denaturation with equal volume of 2 × Sample buffer. The resulting products of Art1 or K63 di-Ub after DUB cleavage were resolved on SDS-PAGE. Art1-HTF proteins were further visualized by immunoblot against FLAG antibody. The DUB cleavage products of K63 di-Ub were strained with Coomassie Blue R250.

## Quantification of Western blot band intensity

Western blot in figures were quantified using ImageJ software. To quantify the degradation of Itr1 and Mup1 proteins, the band densities for the full length Itr1-GFP and Mup1-GFP were quantified using ImageJ. At specific concentration of inositol or methionine, the protein degradation efficiency = (1 − [band density of full length protein after induction]/[band density of full length protein without induction]) × 100%. The significance was determined by two-tail t-test, $\alpha$=0.05 (Bonferroni correction), n=3. n.s., not significant; *, p<0.05; **, p<0.01; ***, p<0.001.

## Quantification of microscopy images

Images of GFP-Rsp5, Art5-GFP, and Art1-mNG were taken by fluorescence microscopy. The fluorescence signal of the target proteins at PM were selected and measured by ImageJ. The corrected total

fluorescence of each selection = selected density – (selected area × mean fluorescence of background readings). The ratio of GFP-Rsp5, Art5-GFP, and Art1-mNG recruitment to PM = (the corrected fluorescence density of the target proteins localized at PM)/(the corrected fluorescence density). The ratios of GFP-Rsp5, Art5-GFP, and Art1-mNG recruitment were measured from n=40 cells.

## Acknowledgements

We appreciate the generous help of Chris Fromme's lab members for reagents, protocols, strains and instruments. We are grateful to Saket Bagde's for his technical assistance of this study. We appreciate Andrew Lombardo for his technical support of Isothermal Titration Calorimetry assay. We are indebted to Jason A MacGurn and Matthew G Baile for critical reading of the manuscript. We thank Hsuan-Chung Ho and Min Wan for reagents and helpful advice. We also thank other members of the Emr lab for helpful discussions. This work was supported by a Cornell University Research Grant (CU563704) to Scott D Emr.

---

## Additional information

### Funding

| Funder | Grant reference number | Author |
| --- | --- | --- |
| Cornell University | CU563704 | Scott D Emr |

The funders had no role in study design, data collection and interpretation, or the decision to submit the work for publication.

### Author contributions

Lu Zhu, Conceptualization, Data curation, Formal analysis, Funding acquisition, Investigation, Methodology, Project administration, Resources, Supervision, Validation, Visualization, Writing – original draft, Writing – review and editing; Qing Zhang, Formal analysis, Investigation, Methodology, Writing – review and editing; Ciro D Cordeiro, Investigation, Writing – review and editing; Sudeep Banjade, Methodology; Richa Sardana, Investigation; Yuxin Mao, Conceptualization, Funding acquisition, Project administration, Resources, Supervision; Scott D Emr, Conceptualization, Funding acquisition, Investigation, Project administration, Resources, Supervision, Writing – review and editing, Methodology

### Author ORCIDs

Lu Zhu http://orcid.org/0000-0001-9438-9296
Sudeep Banjade http://orcid.org/0000-0002-5920-891X
Yuxin Mao http://orcid.org/0000-0002-5064-1397
Scott D Emr http://orcid.org/0000-0002-5408-6781

### Decision letter and Author response

Decision letter https://doi.org/10.7554/eLife.77424.sa1
Author response https://doi.org/10.7554/eLife.77424.sa2

---

## Additional files

### Supplementary files

- Supplementary file 1. All yeast strains and plasmids used in this study.

- Supplementary file 2. Coomassie blue stained SDS-PAGE gels showing the purity of proteins used in the ITC experiment.

- Transparent reporting form

- Source data 1. Source data combined: the uncropped gels or blots for all the figures.

- Source data 2. Source data combined: all the uncropped gels in *Supplementary file 2*.

## Data availability

All data generated or analysed during this study are included in the manuscript. Source Data files have been provided for figures.

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
