## [Editor Report]

This paper provides an important advance in the understanding of the role of Art protein ubiquitylation by Rsp5 in the regulation of cargo import by endocytosis. Rsp5 promotes K63 linked ubiquitylation of Art proteins, thereby enhancing plasma membrane recruitment. Along with detailed biochemical analysis, this study uncovers a novel ubiquitination modification implemented by Rsp5 adaptor proteins, underscoring the regulatory mechanism of how adaptor proteins control the recruitment and activity of Rsp5 for the turnover of membrane proteins.

---

## [Decision Letter]

**Decision letter after peer review:**

Thank you for submitting your article "Adaptor linked K63 di-Ubiquitin activates Nedd4/Rsp5 E3 ligase" for consideration by *eLife*. Your article has been reviewed by 3 peer reviewers, one of whom is a member of our Board of Reviewing Editors, and the evaluation has been overseen by Vivek Malhotra as the Senior Editor. The following individual involved in the review of your submission has agreed to reveal their identity: Danny Huang (Reviewer #3).

Essential revisions:

Overall, the reviewers find the paper to be very interesting and of potential interest to *eLife* readers. Many aspects of the paper are well done. However, some of the conclusions are not sufficiently supported by data. The reviewers feel that among the various aspects that were described in the detailed review, the following four areas need to be addressed.

1) The conclusion that the di-ubiquitin linkage is K63 rests on the use of a K63R mutant. But this could be indirect. The authors should provide further demonstration of linkage type. For example, proteomics analysis or with a K63-specific DUB compared to DUBs specific for other linkage types (UbiCRest).

2) The authors claim that loss of the K63 di-ubiquitination of ART proteins does not affect their interaction with cargoes for ubiquitination and endocytosis. This claim is based on co-IP experiments. However, the western blot signals for Itr1-GFP and Mup1-GFP seem to be over-exposed and saturated in the elution lanes (Figures 2D, E; Figure 2 supp 1D-F), making it difficult to draw meaningful conclusions on the efficiency of the interaction. Along these lines, it is unclear why Art5-K36R and Art5-∆PY are not relocalized to the plasma membrane following inositol treatment (Figure 3B) if the Art5-Itr1 interaction is not affected.

3) The authors conclude that di-Ub binding to Rsp5's exosite stabilizes the ART-Rsp5 interaction. This is supported by ITC experiments with the *S. pombe* orthologs Any1 and Pub1. However, in vivo Art1 and Art5 mutants which cannot be ubiquitinated show a similar interaction with Rsp5 as their wild-type counterparts (Figure 5A-B). Further experiments should be done to demonstrate that di-ubiquitinated ARTs enhance binding to Rsp5 in vivo. For example, the authors could utilize a sequential IP against the ART protein, followed by ubiquitin to demonstrate that the ubiquitinated pool of ARTs preferentially interacts with Rsp5.

4) The proposed model on ubp2 and K48-ubiquitination of ARTs by Rsp5 are seen as preliminary and will require significant work to further justify. In particular, the use of Rsp5 exosite mutant could potentially reduce Rsp5 E3 activity resulting in less ART K63 ubiquitination as well. Given the extent of work needed, the reviewers suggest that it is probably best to tone down the proposed model and revise the text.

*Reviewer #1 (Recommendations for the authors):*

This paper describes an analysis of the role of K63 ubiquitylation of ARTs proteins by Rsp5. ARTs are adaptors for Rsp5 that link it to recruitment to membranes including the plasma membrane where the ubiquitylation activity of Rsp5 promotes cargo endocytosis. Rsp5 di-ubiquitylates several ARTs and this enhances Rsp5 recruitment to membranes and is required for cargo ubiquitylation. Interestingly, the HECT domain exosite blocks the di-K63 ubiquitin from removal by DUBs. When the exosite interaction is not present, di-K63 is removed and ARTs become substrates for K48 ubiquitylation and proteasomal degradation. These studies reveal a detailed mechanism of ARTs modification that couples Rsp5 activity to activation of ARTs for cargo import.

Overall, the studies are well performed and described.

Figure 1. This experiment nicely shows that K364 is the primary site of ubiquitylation in Art5 and that this occurs largely through K63 linkages. For Figure 1C, I found the explanation and conclusion not so clear. The apparent mono-ubiquitylated species seems to still be detected in the K364 mutant – it's just that the abundance of the unmodified form is increased. There could be other weaker sites of ubiquitylation that are modified in the mutant but this isn't really consistent with how this is described. It could also be that when the mono-ubiquitylation cannot be extended, the ubiquitin is removed (which is addressed later in the paper). In supplemental figures, the authors look at Art1 and Art4. The issue described above is clearer with Art4, and somewhat so with Art1. It seems like this inconsistency should be addressed.

For Figure 2, the Art5 K364 mutant has residual processing to yield GFP (Figure 2A), and there is also a small amount of GFP in the vacuole. However, the authors call the "essential". There may be low levels of ubiquitylation of other residues that give this low-level trafficking, but ubiquitylation of that site isn't "essential". The text should be modified.

Figure 3 examines the role of Art ubiquitylation for Rsp5 recruitment to the PM. This is well done and carefully explained.

Figure 4 examines the role of a cargo protein in recruitment, while Figure 5 shows that the interaction between the ligase and the Art protein is enhanced by di-K63 Ub. Overall, the experiments are clearly performed and presented. The use of ITC is a strong aspect of the work. One question is the extent to which potential dimerization of GST might affect interactions.

Figure 6 identified and characterizes a role for Ubp2 in the regulation of Art stability.

The authors propose a model wherein ubiquitylated Art remains associated with Rsp5 exosite. When the exosite is mutated, Ubp2 can remove ubiquitin, which then allows the assembly of K48 chains on Art5 through the same lysine residue, thereby leading to the turnover of Art by the proteasome. What wasn't clear to me is whether the K48 chain assembly is via Rsp5 or a different E3. The model would suggest that it is Rsp5, but I am not sure that the data is presented to demonstrate that.

*Reviewer #2 (Recommendations for the authors):*

1) The authors claim that Art1, Art4, and Art5 are di-ubiquitinated by Rsp5. This conclusion is based on the fact that di-ubiquitination of the three arrestins is dependent on the presence of PY motifs in each protein, and thus their interaction with Rsp5. However, the binding of each ART protein to Rsp5 via its exosite is required to protect the proteins from deubiquitination by Ubp2. Thus, the authors cannot distinguish whether the effect observed by the ∆PY mutants (Figures 1C, Figure 1 supp 1C, and Gid1 supp 2B) are due to ubiquitination by Rsp5 itself or simply to a loss of protection from deubiquitination. Additional experiments would be needed to clarify this point. For example, could the authors reconstitute the ubiquitination reactions in vitro using their purified ART and Rsp5 proteins? The authors could also separate roles of Ub binding to the exosite and other functions by using one of Sidhu's ubiquitin variants that bind the exosite but would not be deconjugated by Ubp2.

2) The conclusion that the di-ubiquitin linkage is K63 rests on the use of a K63R mutant. But this could be indirect. The authors should provide further demonstration of linkage type. For example, proteomics analysis or with a K63-specific DUB compared to DUBs specific for other linkage types (UbiCRest).

3) The authors claim that loss of the K63 di-ubiquitination of ART proteins does not affect their interaction with cargoes for ubiquitination and endocytosis. This claim is based on co-IP experiments. However, the western blot signals for Itr1-GFP and Mup1-GFP are completely over-exposed and saturated in the elution lanes (Figures 2D, E; Figure 2 supp 1D-F), making it impossible to draw meaningful conclusions on the efficiency of the interaction. Along these lines, it is unclear why Art5-K36R and Art5-∆PY are not relocalized to the plasma membrane following inositol treatment (Figure 3B) if the Art5-Itr1 interaction is not affected.

4) The authors conclude that di-Ub binding to Rsp5's exosite stabilizes the ART-Rsp5 interaction. This is supported by ITC experiments with the *S. pombe* orthologs Any1 and Pub1. However, in vivo Art1 and Art5 mutants which cannot be ubiquitinated show a similar interaction with Rsp5 as their wild-type counterparts (Figure 5A-B). Further experiments should be done to demonstrate that di-ubiquitinated ARTs enhance binding to Rsp5 in vivo. For example, the authors could utilize a sequential IP against the ART protein, followed by ubiquitin to demonstrate that the ubiquitinated pool of ARTs preferentially interacts with Rsp5.

*Reviewer #3 (Recommendations for the authors):*

1. The last result section on the involvement of ubp2 on the reversibility of adaptor protein ubiquitination is not clear. Art5 and Art1 did not appear to undergo ubiquitination upon inositol or methionine treatment in δ ubp2 condition (Figure 6 supplement Figure 2A and 2B). Some lanes appear to have more Art5 or Art1 making it difficult to assess the data. This requires quantification.

Figure 6 supplement Figure 2A, there is an extra lane on the upper panel. Please clarify.

2. The authors proposed that ubp2 could reverse K63 ubiquitination of adaptor protein to allow K48-ubiquitination of adaptor protein. Much of this hypothesis was derived using a Rsp5 mutant that is defective in exosite ubiquitin binding. It seems that wild-type Rsp5 protects K63-di-ubiquitin modified Art adaptor protein from deubiquitination, so it is unclear whether K48-ubiquitination occurs under a wild-type setting. Furthermore, the model (Figure 6H) suggests that Rsp5 catalyzes K48-linked ubiquitination of Art adaptor protein but there is little evidence supporting this claim and how Rsp5 switches from K63- to K48-ubiquitination is unclear.

3. The method stated that Any1-K63-di-ubiquitin was produced using Pub1, UbcH5b and K63-di-ubiquitin. Was the modification at a single lysine site? Please provide SDS-PAGE showing the purity Any1-K63-di-ubiquitin and other proteins used in the ITC experiments.

4. The title suggested that adaptor K63-di-ubiquitin activates Rsp5/Nedd4 ligase. However, the data only showed enhanced binding and required for the recruitment to the cargo and ubiquitination. Does Any1-K63-di-ubiquitin enhance the ubiquitin ligase activity of Pub1 or does K63-di-ubiquitin enhance the activity of Rsp5 better than mono-Ub?

5. For ITC experiments, it was difficult to visualize the ITC curves for the controls. Best to show the control curves in the supplemental. Is it possible to estimate the stoichiometry of the binding? The molar ratio for K63-di-ubiquitin and linear-di-ubiquitin do not appear to be 1:1 binding.

6. The linear di-ubiquitin exhibited a high affinity for Rsp5 exosite but Art KR mutant fused with linear di-ubiquitin at the C-terminus did not rescue the defect. Could the authors provide an explanation?

7. Please state how protein concentrations are determined in the method. Is the protein concentration of di-ubiquitin calculated based on dimer or monomer?

---

## [Author Response]

Essential revisions:Overall, the reviewers find the paper to be very interesting and of potential interest to eLife readers. Many aspects of the paper are well done. However, some of the conclusions are not sufficiently supported by data. The reviewers feel that among the various aspects that were described in the detailed review, the following four areas need to be addressed.1) The conclusion that the di-ubiquitin linkage is K63 rests on the use of a K63R mutant. But this could be indirect. The authors should provide further demonstration of linkage type. For example, proteomics analysis or with a K63-specific DUB compared to DUBs specific for other linkage types (UbiCRest).

We performed the UbiCRest assay. We demonstrate that Art1 is ubiquitinated by K63 linked di-Ub chain. We found that ubiquitinated Art1 can be cleaved by the general DUB USP2, K63 specific DUB AMSH and K11, K63 specific DUB Cezanne. Further, the ubiquitinated Art1 is not cleaved by OTUB1 (K48-specific), OTULIN (M1-specific) and YOD1 (K6, K11, K27, K29 or K33 linkage accessible).

2) The authors claim that loss of the K63 di-ubiquitination of ART proteins does not affect their interaction with cargoes for ubiquitination and endocytosis. This claim is based on co-IP experiments. However, the western blot signals for Itr1-GFP and Mup1-GFP seem to be over-exposed and saturated in the elution lanes (Figures 2D, E; Figure 2 supp 1D-F), making it difficult to draw meaningful conclusions on the efficiency of the interaction. Along these lines, it is unclear why Art5-K36R and Art5-∆PY are not relocalized to the plasma membrane following inositol treatment (Figure 3B) if the Art5-Itr1 interaction is not affected.

To resolve the overexposed Mup1 and Itr1 band density in our immunoblots, we reperformed the Co-IP experiments in the Figures 2D, 2E; Figure 2-figure supplement 1D-F.

3) The authors conclude that di-Ub binding to Rsp5's exosite stabilizes the ART-Rsp5 interaction. This is supported by ITC experiments with the *S. pombe* orthologs Any1 and Pub1. However, in vivo Art1 and Art5 mutants which cannot be ubiquitinated show a similar interaction with Rsp5 as their wild-type counterparts (Figure 5A-B). Further experiments should be done to demonstrate that di-ubiquitinated ARTs enhance binding to Rsp5 in vivo. For example, the authors could utilize a sequential IP against the ART protein, followed by ubiquitin to demonstrate that the ubiquitinated pool of ARTs preferentially interacts with Rsp5.

To answer the reviewer’s question to distinguish the Rsp5 binding affinity between ARTs-WT and ARTs-KR mutant, we performed size exclusion chromatography analysis of the Any1–Pub1 complex in the Figure 5-figure supplement 2. The interaction between Rsp5/Nedd4 E3 ligases and the non-ubiquitinated adaptor proteins is quite strong as evidenced by the co-fractionation between Any1 and Pub1 by size exclusion chromatography. It explained why in vivo Art1 and Art5 mutants which cannot be ubiquitinated show a similar interaction with Rsp5 as their wild-type counterparts in a co-IP assay, which usually has a low sensitivity for detecting protein-protein interactions. However, using the more sensitive and quantitative ITC approach (Figure 5E-G), we were able to observe a higher affinity between wild type E3 and the di-Ub-adaptor compared to the non-ubiquitinated adaptor or exosite mutant E3 with the di-Ub-adaptor. These results allowed us to conclude that di-Ub binding to Rsp5's exosite stabilizes the ART-Rsp5 interaction.

4) The proposed model on ubp2 and K48-ubiquitination of ARTs by Rsp5 are seen as preliminary and will require significant work to further justify. In particular, the use of Rsp5 exosite mutant could potentially reduce Rsp5 E3 activity resulting in less ART K63 ubiquitination as well. Given the extent of work needed, the reviewers suggest that it is probably best to tone down the proposed model and revise the text.

For the proposed model on ubp2 and K48-ubiquitination of ARTs by Rsp5, we removed the working model and revised the text.

Reviewer #1 (Recommendations for the authors):This paper describes an analysis of the role of K63 ubiquitylation of ARTs proteins by Rsp5. ARTs are adaptors for Rsp5 that link it to recruitment to membranes including the plasma membrane where the ubiquitylation activity of Rsp5 promotes cargo endocytosis. Rsp5 di-ubiquitylates several ARTs and this enhances Rsp5 recruitment to membranes and is required for cargo ubiquitylation. Interestingly, the HECT domain exosite blocks the di-K63 ubiquitin from removal by DUBs. When the exosite interaction is not present, di-K63 is removed and ARTs become substrates for K48 ubiquitylation and proteasomal degradation. These studies reveal a detailed mechanism of ARTs modification that couples Rsp5 activity to activation of ARTs for cargo import.Overall, the studies are well performed and described.Figure 1. This experiment nicely shows that K364 is the primary site of ubiquitylation in Art5 and that this occurs largely through K63 linkages. For Figure 1C, I found the explanation and conclusion not so clear. The apparent mono-ubiquitylated species seems to still be detected in the K364 mutant – it's just that the abundance of the unmodified form is increased. There could be other weaker sites of ubiquitylation that are modified in the mutant but this isn't really consistent with how this is described. It could also be that when the mono-ubiquitylation cannot be extended, the ubiquitin is removed (which is addressed later in the paper). In supplemental figures, the authors look at Art1 and Art4. The issue described above is clearer with Art4, and somewhat so with Art1. It seems like this inconsistency should be addressed.

We rephrased text. We also noticed the difference between Art4 and Art5 (and Art1). Probably, there are multiple ubiquitination sites in Art5 and Art1 and the K486 of Art1 and K364 of Art5 are the primary ubiquitination sites. The ubiquitination site mutant of Art4 used in the figure 1—figure supplemental 2 is a quadruple mutant (K235R/K245R/K264R/K267R), reported by Becuwe and his coworkers (Becuwe et al., 2012, PMID: 22249293). The ubiquitination of Art4 is largely ablated in this quadruple mutant.

For Figure 2, the Art5 K364 mutant has residual processing to yield GFP (Figure 2A), and there is also a small amount of GFP in the vacuole. However, the authors call the "essential". There may be low levels of ubiquitylation of other residues that give this low-level trafficking, but ubiquitylation of that site isn't "essential". The text should be modified.

We revised the text.

Figure 3 examines the role of Art ubiquitylation for Rsp5 recruitment to the PM. This is well done and carefully explained.

Thanks!

Figure 4 examines the role of a cargo protein in recruitment, while Figure 5 shows that the interaction between the ligase and the Art protein is enhanced by di-K63 Ub. Overall, the experiments are clearly performed and presented. The use of ITC is a strong aspect of the work. One question is the extent to which potential dimerization of GST might affect interactions.

First of all, in this pull-down assay, GST fusion proteins were immobilized on the Glutathione Sepharose resin. The fused baits (HECT domain and its mutants) will form a surface layer of protein with a high concentration around the resin. In this experiment, we have appropriate GST fusion controls to show only the wild-type HECT domain was able to pull down Ub chains but not the designed points mutants (Figure 4A).

Second, the potential dimerization of GST will unlikely affect the interactions between Ub or linear forms of Ub chain variants and HECT domain, or the overall distribution of the fused baits on the surface of the resin. The dimeric nature of the traditional GST-tag could be a problem only if the proteins fused to GST are dimers themselves. It was reported that free ubiquitin dimerizes noncovalently in solution with a dissociation constant Kd of (4.9±0.3) mM (Liu, Tang, et al., Angew Chem Int Ed Engl, 2012. PMID: 22109817). For our GST-pull down assays in this study, our GST-fusion proteins are as low as 2µM in the binding reaction mixture, in which Ub will mainly stay in monomeric form in vast majority.

Figure 6 identified and characterizes a role for Ubp2 in the regulation of Art stability.The authors propose a model wherein ubiquitylated Art remains associated with Rsp5 exosite. When the exosite is mutated, Ubp2 can remove ubiquitin, which then allows the assembly of K48 chains on Art5 through the same lysine residue, thereby leading to the turnover of Art by the proteasome. What wasn't clear to me is whether the K48 chain assembly is via Rsp5 or a different E3. The model would suggest that it is Rsp5, but I am not sure that the data is presented to demonstrate that.

The Art5 degradation in an exosite mutant can be rescued by PY mutant, suggesting that Rsp5-Art5 interaction is required for the Art5 ubiquitination.

Reviewer #2 (Recommendations for the authors):1) The authors claim that Art1, Art4, and Art5 are di-ubiquitinated by Rsp5. This conclusion is based on the fact that di-ubiquitination of the three arrestins is dependent on the presence of PY motifs in each protein, and thus their interaction with Rsp5. However, the binding of each ART protein to Rsp5 via its exosite is required to protect the proteins from deubiquitination by Ubp2. Thus, the authors cannot distinguish whether the effect observed by the ∆PY mutants (Figures 1C, Figure 1 supp 1C, and Gid1 supp 2B) are due to ubiquitination by Rsp5 itself or simply to a loss of protection from deubiquitination. Additional experiments would be needed to clarify this point. For example, could the authors reconstitute the ubiquitination reactions in vitro using their purified ART and Rsp5 proteins? The authors could also separate roles of Ub binding to the exosite and other functions by using one of Sidhu's ubiquitin variants that bind the exosite but would not be deconjugated by Ubp2.

a. The deubiquitination by Ubp2 can be distinguished from the ΔPY mutant. First, the

PY motif and WW domain interaction is essential for binding of adaptor proteins

with Rsp5 E3 ligase. No ubiquitination occurs in ΔPY mutants. Second, the

remaining non-ubiquitinated Art5 in the exosite mutant resembles the amount of

the non- ubiquitinated form of Art5 in the WT-Rsp5 condition, whereas art5-DPY

levels are dramatically enhanced for the non-ubiquitinated form of Art5. This is

because the art5-ΔPY loses the interaction with Rsp5 and cannot be ubiquitinated

and degraded.

b. We performed the in-vitro ubiquitination experiment. See Author response image 1. In the figure A, the Any1 can be poly-ubiquitinated by E3 ligase Pub1 in the presence of WT-Ub or Ub-63K-only (lane 2 and 4). This polyubiquitination of Any1 can be reversed in the Ub-K0 or Ub-K63R mutants. In the figure B, we showed that the polyubiquitinated

**Author response image 1. sa2fig1:** Any1 is ubiquitinated in K63 linkage using in-vitro assay. (A) E1 (Uba1), E2 (UbcH5), E3 (Pub1), substrate (Any1) and ubiquitin variants were mixed together in the 1x PBDM buffer. (B) E1 (Uba1), E2 (UbcH5), E3 (Pub1), substrate (Any1), WT-Ub, Ubp2 and the adaptor protein Ucp6 in different combination were mixed together in the 1x PBDM buffer. Thereactions were incubated in the room temperature for 30minutes then quenched by 2x sample buffer. The resulting products were solved on 10% SDS-PAGE gel then stained with Coomassie Blue R-250.

Any1 by Pub1 can be cleaved by Ubp2 and only Any1-Ub and Any1-2Ub (or Any1-diUb) can be protected. Together, these results suggest that Any1 can be ubiquitinated by Pub1 with K63 linked Ub chain, which can be further cleaved down to 2xUb by Ubp2. However, we believe that the in-vivo results are more relevant for the current study and these results will not be included in the paper.

2) The conclusion that the di-ubiquitin linkage is K63 rests on the use of a K63R mutant. But this could be indirect. The authors should provide further demonstration of linkage type. For example, proteomics analysis or with a K63-specific DUB compared to DUBs specific for other linkage types (UbiCRest).

We performed the UbiCRest assay as advised by the reviewer. Please see Figure 1—figure supplement 3.

3) The authors claim that loss of the K63 di-ubiquitination of ART proteins does not affect their interaction with cargoes for ubiquitination and endocytosis. This claim is based on co-IP experiments. However, the western blot signals for Itr1-GFP and Mup1-GFP are completely over-exposed and saturated in the elution lanes (Figures 2D, E; Figure 2 supp 1D-F), making it impossible to draw meaningful conclusions on the efficiency of the interaction. Along these lines, it is unclear why Art5-K36R and Art5-∆PY are not relocalized to the plasma membrane following inositol treatment (Figure 3B) if the Art5-Itr1 interaction is not affected.

a. We re-performed the Co-IP experiments in the figure 2D, 2D, figure 2—figure supplement 1D-1F.

b. We have similar observations indicating that Art1 ubiquitination and PY motifs are required for methionine induced PM recruitment of Art1. Probably, the interaction between the adaptors and cargo proteins is required, but not sufficient, for adaptor protein PM recruitment. Rsp5 is recruited by the adaptor to the target membrane and the interaction between Rsp5 and adaptors enhances adaptor recruitment to PM.

4) The authors conclude that di-Ub binding to Rsp5's exosite stabilizes the ART-Rsp5 interaction. This is supported by ITC experiments with the *S. pombe* orthologs Any1 and Pub1. However, in vivo Art1 and Art5 mutants which cannot be ubiquitinated show a similar interaction with Rsp5 as their wild-type counterparts (Figure 5A-B). Further experiments should be done to demonstrate that di-ubiquitinated ARTs enhance binding to Rsp5 in vivo. For example, the authors could utilize a sequential IP against the ART protein, followed by ubiquitin to demonstrate that the ubiquitinated pool of ARTs preferentially interacts with Rsp5.

a. To answer the reviewer’s question, we performed size exclusion chromatography analysis of the Any1–Pub1 complex in the Figure 5—figure supplement 2. The interaction between Rsp5/Nedd4 E3 ligases and the non-ubiquitinated adaptor proteins is quite strong as evidenced by the co-fractionation between Any1 and Pub1 by size exclusion chromatography. It explained why in vivo Art1 and Art5 mutants which cannot be ubiquitinated show a similar interaction with Rsp5 as their wild-type counterparts in a co-IP assay, which usually has a low sensitivity for detecting protein-protein interactions.

b. Thanks for the suggestion. We do not think the sequential Ubiquitin pull down is suitable for this experiment. We assume that ARTs-WT can be pulled down in the sequential IP against Ub. Due to the lack of ubiquitination in the ARTs-KR mutant, the sequential IP of ARTs-KR mutant using Ub as bait will not yield any signal in the ARTs-KR mutant. If the output of ARTs-WT and ARTs-KR mutant are not equal, we cannot compare the amount of co-immunoprecipitated Rsp5. However, using the more sensitive and quantitative ITC approach (Figure 5E-G), we were able to observe a higher affinity between wild type E3 and the di-Ub-adaptor compared to the non-ubiquitinated adaptor or exosite mutant E3 with the di-Ub-adaptor. These results allowed us to conclude that di-Ub binding to Rsp5's exosite stabilizes the ART-Rsp5 interaction.

Reviewer #3 (Recommendations for the authors):1. The last result section on the involvement of ubp2 on the reversibility of adaptor protein ubiquitination is not clear. Art5 and Art1 did not appear to undergo ubiquitination upon inositol or methionine treatment in δ ubp2 condition (Figure 6 supplement Figure 2A and 2B). Some lanes appear to have more Art5 or Art1 making it difficult to assess the data. This requires quantification.

We added the quantification of the data in the Figure 6—figure supplement 2.

Figure 6 supplement Figure 2A, there is an extra lane on the upper panel. Please clarify.

Thanks. The 7^th^ lane was included in the figure by mistake.

2. The authors proposed that ubp2 could reverse K63 ubiquitination of adaptor protein to allow K48-ubiquitination of adaptor protein. Much of this hypothesis was derived using a Rsp5 mutant that is defective in exosite ubiquitin binding. It seems that wild-type Rsp5 protects K63-di-ubiquitin modified Art adaptor protein from deubiquitination, so it is unclear whether K48-ubiquitination occurs under a wild-type setting. Furthermore, the model (Figure 6H) suggests that Rsp5 catalyzes K48-linked ubiquitination of Art adaptor protein but there is little evidence supporting this claim and how Rsp5 switches from K63- to K48-ubiquitination is unclear.

We removed the working models and toned down the text.

3. The method stated that Any1-K63-di-ubiquitin was produced using Pub1, UbcH5b and K63-di-ubiquitin. Was the modification at a single lysine site? Please provide SDS-PAGE showing the purity Any1-K63-di-ubiquitin and other proteins used in the ITC experiments.

Any1 is mainly ubiquitinated at K263 residue (Nakashima, Kikkawa, et al., Biol Open. 2014. PMID: 24876389). The proteins used in the ITC experiment are shown in the Supplementary file 2.

4. The title suggested that adaptor K63-di-ubiquitin activates Rsp5/Nedd4 ligase. However, the data only showed enhanced binding and required for the recruitment to the cargo and ubiquitination. Does Any1-K63-di-ubiquitin enhance the ubiquitin ligase activity of Pub1 or does K63-di-ubiquitin enhance the activity of Rsp5 better than mono-Ub?

a. Nedd4/Rsp5 family E3 ligases mediate numerous cellular processes. Under environmental cues, adaptor proteins of Nedd4/Rsp5 E3 ligases provide a cargospecific quality-control pathway that mediates endocytic downregulation by coupling Rsp5/Nedd4 to diverse plasma membrane proteins. Many Nedd4 family E3 ligase adaptors were self-ubiquitinated for activation of the E3 ligase. The mechanism for how adaptor protein ubiquitination activates Rsp5/Nedd4 E3 ligase has been a long-standing fundamental question in the field. For the first time, our study identified that the adaptor proteins ARTs undergo di-Ub modification, which strengthens the binding with Rsp5 and promotes the efficient recruitment of Rsp5. Adaptor linked di-Ub will enhance the recruitment of Rsp5 locally for specific cargo protein ubiquitination reactions but is unlikely change the Rsp5 ligase activity. The Nedd4/Rsp5 exosite is critical for the HECT domain to bind Ub moieties but exosite mutations do not alter the E3 ligase catalytic activity (Maspero, Polo, et al., EMBO Rep. 2011. PMID: 21399620). This result supports a model that the in-trans interaction between the HECT domain N-lobe region and di-Ub does not interfere the HECT domain catalytic activity.

b. Yes. Based on our recent in-vitro reconstituted ubiquitination assay, we found that K63-di-ubiquitin enhances cargo protein ubiquitination better than mono-Ub (unpublished). We found it quite interesting, and the results fit our current working model. Since this result is beyond the scope of this study, we do not discuss it in the manuscript.

5. For ITC experiments, it was difficult to visualize the ITC curves for the controls. Best to show the control curves in the supplemental. Is it possible to estimate the stoichiometry of the binding? The molar ratio for K63-di-ubiquitin and linear-di-ubiquitin do not appear to be 1:1 binding.

a. The control curves are now shown in the supplemental figures separately.

b. Thanks for your comments. We also noticed that molar ratio for K63-di-ubiquitin and linear-di-ubiquitin do not appear to be 1:1 binding. It seems like multiple sites in the HECT domain N-lobe region were involved in Ub binding. This is in line with the previous published results by French and coworkers (French, Hicke, et al., JBC. 2009. PMID: 19252184). Three N-lobe mutations (Y516A, F618A, and V621A/V622A) of Rsp5 completely abolished ubiquitin binding. Three mutations (N513A, Y521A, and R651A) of Rsp5 caused a reduction in binding with Ub. Rsp5(N534A) reproducibly enhanced Ub binding <inline-graphic mimetype="image" mime-subtype="png" xlink:href="media/image1.png" />2–3-fold relative to wild-type. A highresolution atomic structure of the ART family of adaptor proteins in complex with Rsp5 will be required to assign the binding stoichiometry for how multiple Ubbinding sites in the HECT domain interact with a di-Ub molecule and how the diubiquitinated adaptors engage the HECT E3 ligases.

6. The linear di-ubiquitin exhibited a high affinity for Rsp5 exosite but Art KR mutant fused with linear di-ubiquitin at the C-terminus did not rescue the defect. Could the authors provide an explanation?

These results suggest that di-ubiquitin needs to be conjugated at specific residues for proper functionality. It also implies that Need4/Rsp5 E3 ligases need to be engaged with di-ubiquitinated adaptors precisely to direct the E3 ligase activity to the appropriate cargo proteins.

7. Please state how protein concentrations are determined in the method. Is the protein concentration of di-ubiquitin calculated based on dimer or monomer?

a. Protein concentrations were determined by absorbance at 280 nm (divided by extinction coefficient for specific proteins).

b. The di-Ub concentrations were calculated based on the extinction coefficient of the ubiquitin dimer.